# Expanding the Capability Frontier of LLM Agents with ZPD-Guided Data Synthesis

**Xuanzhong Chen[1,][*], Zile Qiao[2,][*,][†], Guoxin Chen[2], Liangcai Su[2], Zhen Zhang[2],**
**Xinyu Wang[2], Pengjun Xie[2], Fei Huang[2], Jingren Zhou[2], Yong Jiang[2,][†], Ting Chen[1,][†]**

[1] Department of Computer Science and Technology, Tsinghua University
[2] Tongyi Lab, Alibaba Group

cxz23@mails.tsinghua.edu.cn,{qiaozile.qzl, yongjiang.jy}@alibaba-inc.com

## Abstract

Unlocking advanced reasoning in large language model agents is hindered by a scarcity of training data situated at the very frontier of their capabilities. We address this with a novel data synthesis approach inspired by the educational theory of the `Zone of Proximal Development` (ZPD), which conceptualizes this frontier as tasks an LLM cannot solve independently but can master with guidance. We operationalize this principle through the **AgentFrontier Data Engine**, an automated pipeline that synthesizes high-quality, multidisciplinary data situated precisely within an LLM's ZPD. The engine yields two synergistic outputs: knowledge-intensive data for continued pre-training and frontier-level reasoning trajectories for post-training. Concurrently, it produces the **ZPD Exam**, a self-evolving benchmark for evaluating agent capabilities by compelling them to reason beyond their parameterized knowledge. By training our **AgentFrontier-30B-A3B** model on the synthesized data, we achieve state-of-the-art results on demanding benchmarks like `Humanity's Last Exam`, outperforming several leading proprietary agents. This work establishes ZPD-guided data synthesis as a scalable and effective paradigm for cultivating increasingly capable LLM agents.

## 1 Introduction

While large language models (LLMs) have demonstrated impressive proficiency on various fundamental reasoning tasks (Rein et al., 2023; Wang et al., 2024; Tian et al., 2024), they still struggle with the scenarios demanding in-depth, cross-domain, and integrative reasoning (Mialon et al., 2023; Wei et al., 2025; Phan et al., 2025). This gap presents a critical impediment in the pursuit of artificial general intelligence (AGI). Achieving such a leap necessitates a paradigm shift from reliance on static, internal knowledge to dynamic, agentic capabilities such as tool using (Qin et al., 2024), self-reflection (Shinn et al., 2023), iterative planning, and multi-step reasoning. However, the development of such agentic skills is hampered by a dual challenge: a scarcity of training corpora that systematically cultivate these abilities in a unified manner (Shi et al., 2025), and the saturation of existing benchmarks. While expert-crafted evaluations such as *Humanity's Last Exam* (Phan et al., 2025) offer invaluable benchmarks, their prohibitive cost and limited scalability underscore the urgent need for automated pipelines capable of synthesizing frontier-level reasoning tasks.

Recent datasets have significantly enhanced LLMs' single-step reasoning (Liu et al., 2025), but they fall short of targeting the deeper challenge of **knowledge fusion** (Wan et al., 2024): integrating and transforming information across diverse sources. While retrieval-augmented generation (RAG) (Lewis et al., 2020) excels when the answer can be grounded in a single document, its performance degrades on tasks requiring reasoning across heterogeneous information. This deficiency traces back to the dominant data-synthesis paradigms, which fall into two broad categories: query-centric methods (Yan et al., 2025) that generate variations of existing question–answer (QA) pairs, and document-centric methods (Fan et al., 2025; Yuan et al., 2025) that derive document-grounded QA pairs from the corpus. Both approaches primarily assess localized comprehension, akin to examining a student on individual textbook chapter rather than their ability to synthesize insights across an

---

[*] Equal contribution. [†] Corresponding authors.

entire curriculum. In contrast, complex real-world tasks such as academic research, legal analysis, or engineering design demand multi-document synthesis and cross-domain knowledge fusion. Human experts rarely treat information in isolation; instead, they connect, contrast, and integrate it to derive in-depth insights, which is the intrinsic essence of **deep research** (OpenAI, 2025a; Google, 2025). Cultivating such synthetic reasoning capacity is paramount to advancing LLMs toward higher forms of intelligence.

The central challenge of effective data synthesis lies not in merely generating difficult tasks, but in precisely calibrating them to the frontier of a model's competence: complex enough to exceed the boundary of the model's intrinsic capability, yet solvable with appropriate support. Existing approaches typically rely on coarse-grained difficulty annotations (Su et al., 2025) or heuristically stacked constraints (Patel et al., 2025), lacking a precise mechanism for targeting this frontier. In practice, self-generated approaches tend to yield data confined within the model's own expressive ceiling, thus hindering systematic difficulty progression. To address this, we draw inspiration from the educational psychology concept of the *Zone of Proximal Development* (ZPD) (Vygotsky, 1978; McLeod, 2012), which defines the cognitive space where a learner can succeed with guidance on tasks they cannot solve alone. We operationalize this by defining two personas: the **Less Knowledgeable Peer** (LKP), a base LLM, and the **More Knowledge-**

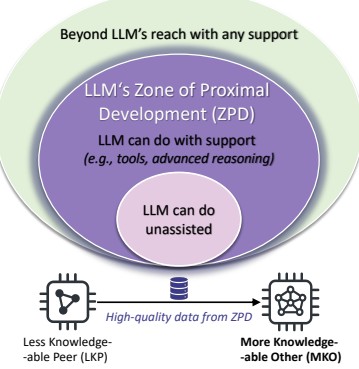

Figure 1: High-quality data within an LLM's ZPD catalyzes its transformation from LKP to MKO.

**able Other** (MKO), a superior tool-augmented agent with advanced reasoning. By definition, tasks that the LKP fails but the MKO masters are situated within the model's ZPD. This provides a principled mechanism for identifying maximally informative training resources and, crucially, allows for a continuously adaptive curriculum that evolves as the model's own capability frontier expands.

We instantiate this principle in the **AgentFrontier Engine**, a novel data synthesis framework designed to generate complex-reasoning data within LLM's ZPD. The engine operates via a process of adversarial calibration, dynamically probing the capability frontier of the LLMs. It constructs multi-disciplinary QA pairs that necessitate knowledge fusion across multiple web sources, moving beyond simple fact retrieval. Our engine employs a dual-pipeline architecture: tasks solvable by the LKP are curated as knowledge-intensive data for continued pre-training (CPT), while those requiring the MKO are designated as frontier-level data for post-training. This design establishes a virtuous cycle, yielding a continuous stream of adaptive data that propels model capability forward.

Our contributions are threefold:

1. We present **AgentFrontier Engine**, a scalable data synthesis framework founded on the theory of *Zone of Proximal Development* (ZPD). By integrating agentic refinement and LKP–MKO adversarial calibration, it generates a dual stream of knowledge-intensive data for broad competence and frontier-level data for advanced reasoning.

2. We establish **ZPD Exam**, an automated benchmark designed to probe the ZPD of LLMs. It assesses advanced capabilities such as tool using and in-depth reasoning by complex multi-disciplinary questions that require cross-document knowledge fusion and deep research.

3. We demonstrate the effectiveness of our framework by developing **AgentFrontier-30B-A3B**. By applying continued pre-training on 50 billion tokens of knowledge-intensive data and fine-tuning on 12,000 frontier-level trajectories synthesized by our engine, the resulting model achieves 28.6% on the challenging HLE benchmark and sets state-of-the-art performance on ZPD Exam-v1, R-Bench-T and xBench-ScienceQA.

## 2 AGENTFRONTIER DATA ENGINE

**AgentFrontier Engine** addresses the critical need for training data that fosters knowledge fusion and complex reasoning, which operationalizes the theoretical framework of the *Zone of Proximal Development* to generate challenging tasks that reside at the frontier of a LLM's capabilities. Instead of passively curating existing information, the engine is designed to actively forge complexity

through a three-stage agentic synthesis pipeline. This process aims to evolve LLMs from knowledge retrievers into sophisticated reasoning agents. The entire workflow, depicted in Figure 2, transforms a raw document corpus $\mathcal{C}_{\text{raw}}$ into a calibrated, high-value dataset $\mathcal{D}_{\text{ZPD}}$. The detailed procedure is presented in Algorithm 1. The detailed prompts are provided in the Appendix F.

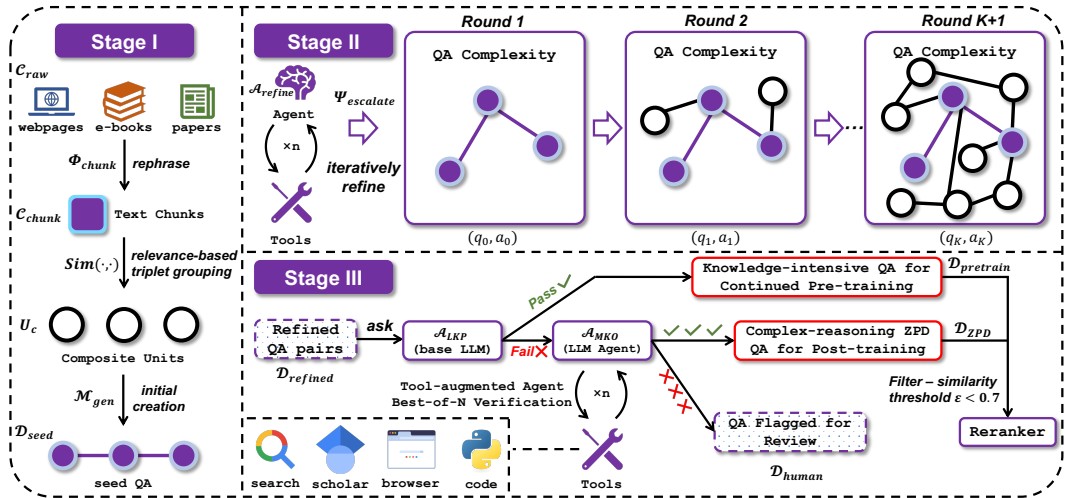

Figure 2: The three-stage pipeline of the AgentFrontier Engine. Stage I generates seed QA pairs from multiple sources. Stage II iteratively escalates their complexity using a tool-augmented agent. Stage III applies a ZPD-based calibration filter to isolate high-value training samples.

## 2.1 STAGE I: SEED QUESTION GENERATION FOR KNOWLEDGE FUSION

The pipeline begins with a diverse, multi-disciplinary corpus $\mathcal{C}_{\text{raw}}$ of one million public documents. We first employ a powerful LLM, Qwen3-235B-A22B (Yang et al., 2025), as a chunking function $\Phi_{\text{chunk}}$ to preprocess the corpus. This function cleans artifacts (e.g., HTML tags) and condenses long texts into information-dense chunks $\mathcal{C}_{\text{chunk}}$, such that $\mathcal{C}_{\text{chunk}} = \bigcup_{d \in \mathcal{C}_{\text{raw}}} \Phi_{\text{chunk}}(d)$.

To generate tasks that inherently demand knowledge fusion, we synthesize questions from **composite units**—groups of thematically related chunks. To overcome the computational infeasibility of a combinatorial search, we adopt an efficient, retrieval-based approach. We first build a vector index over $\mathcal{C}_{\text{chunk}}$ and, for each chunk $c_i$, retrieve its $k_{\text{nn}}$ nearest neighbors. Within this local neighborhood, we search for triplets $(c_i, c_j, c_k)$ that exhibit high thematic coherence, formally defined as $\text{Sim}(c_x, c_y) > \tau_{\text{theme}}$ for all distinct pairs, where $\text{Sim}(\cdot, \cdot)$ is a semantic similarity function.

These composite units are then fed to a generator model, $\mathcal{M}_{\text{gen}}$ (DeepSeek-R1-0528 (Guo et al., 2025a)), to synthesize initial question-answer pairs. This process yields a seed dataset that serves as the foundation for complexity escalation: $\mathcal{D}_{\text{seed}} = \{(q_0, a_0) = \mathcal{M}_{\text{gen}}(U_c) \mid U_c \text{ is a composite unit}\}$.

## 2.2 STAGE II: ESCALATING COMPLEXITY THROUGH AGENTIC REFINEMENT

The core of our engine is an iterative refinement loop driven by a refinement agent $\mathcal{A}_{\text{refine}}$, which integrates DeepSeek-R1 with a tool suite $\mathcal{T} = \{T_{\text{search}}, T_{\text{scholar}}, T_{\text{browser}}, T_{\text{code}}\}$. For a QA pair $(q_k, a_k)$ at iteration $k$, the agent applies an escalation operator $\Psi_{\text{escalate}}$ to generate a more sophisticated pair $(q_{k+1}, a_{k+1}) = \Psi_{\text{escalate}}(q_k, a_k, \mathcal{A}_{\text{refine}})$. This operator enriches the QA along four dimensions:

- **Knowledge Expansion:** It actively queries external sources to retrieve and weave in relevant background knowledge, broadening the informational scope of the question.
- **Conceptual Abstraction:** It conducts in-depth analysis of the core concepts within the provided materials, abstracting higher-level principles or identifying subtle relationships.
- **Factual Grounding:** It performs multi-source cross-validation and targeted augmentation to enhance the factual accuracy and depth of the content.
- **Computational Formulation:** It leverages the Python execution to craft QA that require quantitative calculation or logical simulation, assessing reasoning and computational skills.

This self-bootstrapping process creates a virtuous cycle, where the output of one iteration becomes the input for the next, building increasingly more intricate reasoning paths. Figure 6 illustrates an example where a question is progressively refined by interleaving web search with numerical computation. After $K$ iterations, this stage produces a dataset of highly complex QA pairs, $\mathcal{D}_{\text{refined}}$.

## 2.3 STAGE III: ZPD-BASED FILTERING AND CALIBRATION

Not all synthesized QA pairs are equally valuable for training. To isolate tasks that reside precisely within an LLM's ZPD, we introduce a rigorous calibration mechanism based on our **LKP-MKO** framework. We instantiate a **Less Knowledgeable Peer** ($\mathcal{A}_{\text{LKP}}$) with the base DeepSeek-R1-0528 model (without tools) (Guo et al., 2025a) and a **More Knowledgeable Other** ($\mathcal{A}_{\text{MKO}}$) with the powerful, tool-augmented DeepSeek-V3.1 agent (Liu et al., 2024).

For each candidate pair $(q, a) \in \mathcal{D}_{\text{refined}}$, we first assess its difficulty. Let IsSolvableBy$(\mathcal{A}, q, a) \in \{0, 1\}$ be a binary function, implemented by an automated judge (GPT-4o (OpenAI, 2024)), which returns 1 if agent $\mathcal{A}$ correctly answers $q$. (a) If IsSolvableBy$(\mathcal{A}_{\text{LKP}}, q, a) = 1$, the pair is deemed too simple and is allocated to a general knowledge dataset $\mathcal{D}_{\text{pretrain}}$ for continued pre-training. (b) If IsSolvableBy$(\mathcal{A}_{\text{LKP}}, q, a) = 0$, the pair is challenging and passed to the MKO for further evaluation.

To stratify the challenging data, $\mathcal{A}_{\text{MKO}}$ performs Best-of-N (BoN) verification with $N = 3$, generating $N$ independent solutions $\{s_1, \ldots, s_N\}$. The data is then partitioned based on the outcome:

- **Verified for Post-Training ($\mathcal{D}_{\text{ZPD}}$):** If the MKO finds at least one correct solution (i.e., $\sum_{i=1}^{N} \text{IsCorrect}(s_i, a) \geq 1$), the pair is considered to be within the model's ZPD—challenging yet learnable. These verified pairs form our final training set.
- **Flagged for Human Review ($\mathcal{D}_{\text{human}}$):** If the MKO fails in all $N$ attempts (i.e., $\sum_{i=1}^{N} \text{IsCorrect}(s_i, a) = 0$), the pair is either flawed or exceptionally difficult and is routed to human experts for analysis. The human review process is detailed in the Appendix C.4.

Finally, to ensure dataset diversity, we apply a semantic redundancy filter. A newly generated pair $(q', a')$ is discarded if its question $q'$ is too similar to any question already in $\mathcal{D}_{\text{ZPD}}$. Specifically, we discard $(q', a')$ if $\max_{(q,a) \in \mathcal{D}_{\text{ZPD}}} \text{Sim}(q', q) \geq \epsilon$, where $\text{Sim}(\cdot, \cdot)$ is measured by a reranker model (Zhang et al., 2025) and the threshold $\epsilon$ is set to 0.7.

Through this three-stage pipeline, the AgentFrontier Engine provides a scalable method for generating complex reasoning data, continuously pushing the boundaries of LLM capabilities.

# 3 ZPD EXAM: A SELF-EVOLVING BENCHMARK FOR LLM AGENTS

Evaluating rapidly advancing LLMs requires benchmarks that co-evolve with their capabilities. While expert-crafted exams like Humanity's Last Exam (Phan et al., 2025) probe the frontier of human knowledge, their static nature and prohibitive creation costs hinder scalable and continuous assessment. We introduce the **ZPD Exam**, an automated and continuously evolving benchmark designed to assess the deep research capabilities of advanced LLM agents.

## 3.1 BENCHMARK CONSTRUCTION: FROM FRONTIER KNOWLEDGE TO AGENTIC RESEARCH

The ZPD Exam is designed to simulate scientific discovery by generating tasks that are intractable using only parametric knowledge, thus compelling models to function as research agents. The benchmark is constructed using our AgentFrontier Engine (Section 2), specifically configured to generate novel, multi-disciplinary questions. Crucially, this benchmark corpus is strictly disjoint from the corpus used to construct our training data, ensuring a fair and uncontaminated evaluation.

**Grounding in the Knowledge Frontier.** We ground this exam in the knowledge frontier by curating a corpus of 30,000 recent scientific papers published between 2023 and 2025, spanning multi-disciplinary domains such as mathematics, computer science, and physics. This ensures that success demands genuine, on-the-fly reasoning and information synthesis, not merely knowledge retrieval.

**Calibrating Tasks to the LLM's ZPD.** From our initial corpus, the AgentFrontier Engine generates candidate questions, which are then subjected to a strict adversarial filter to align with the ZPD of

a baseline model (DeepSeek-R1-0528 (Guo et al., 2025a)). To be included in ZPD Exam-v1, a problem must satisfy a dual constraint: it must be unsolvable by the baseline model in three unaided attempts, yet consistently solvable by the same model across three attempts when granted access to tools. This process isolates problems that are difficult but solvable with assistance, defining the empirical boundary of the model's ZPD.

This automated pipeline enables a flywheel-like iterative process: as models improve, the ZPD exam can be regenerated to target the new frontier, making it a **living benchmark** resistant to saturation. After multiple rounds of validation and deduplication, ZPD Exam-v1 was constructed by sampling 1,024 public questions and a corresponding private set. All questions are open-ended short-answer format, facilitating automated grading. The benchmark composition is detailed in Figure 7.

### 3.2 ZPD Exam: A Diagnostic Benchmark for Agentic Reasoning

The ZPD Exam proposes a new evaluative framework, shifting the focus from an LLM's static parametric knowledge (Hendrycks et al., 2021) to its dynamic capacity for knowledge discovery, which functions as an "open-book" examination where agent must first author the "book" through active exploration and tool use. This design philosophy deliberately situates the challenges within the ZPD for current LLMs, a calibration confirmed by their low initial scores (Figure 3). Our empirical results validate this diagnostic power, revealing a clear stratification of agent performance into three distinct zones.

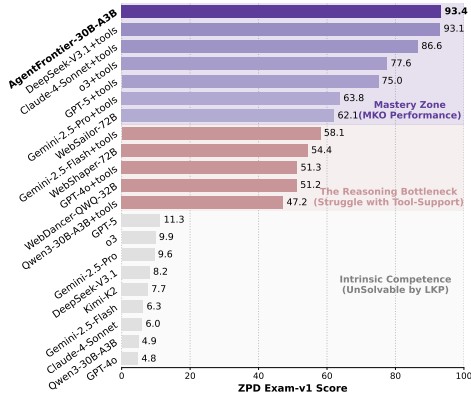

Figure 3: Performance of LLM agents on the ZPD Exam-v1 benchmark, stratified into three distinct capability zones.

**Zone 1: Intrinsic Competence (Score < 20).** This tier establishes the baseline, reflecting the performance of LLMs relying solely on their parametric knowledge (e.g., GPT-5 and Gemini-2.5-Pro without tools). By design, the problems are intractable without external information, confirming that these tasks lie outside the models' unaided capabilities. This zone effectively establishes a baseline, quantifying the limits of intrinsic, "closed-book" reasoning, confirming that any score above this threshold is directly attributable to the agent's ability to leverage external tools support.

**Zone 2: The Reasoning Bottleneck (Score 20-60).** This intermediate tier characterizes the ZPD itself, where agents (e.g., GPT-4o with tools, WebShaper-72b) can achieve partial success with assistance but lack mastery. This zone highlights the benchmark's crucial distinction from standard RAG evaluations. While RAG tests comprehension of a given context, agents here falter in the more demanding task of autonomously discovering, structuring, and reasoning over the necessary information. Their failures stem not from tool-level errors but from a higher-order "reasoning bottleneck": a deficit in strategic planning, synthesizing information across multiple tool calls, and adapting their approach. This reveals that access to tools is necessary but insufficient; the primary limiting factor is the agent's meta-cognitive ability to orchestrate these tools effectively.

**Zone 3: Emergent Mastery (Score > 60).** Agents in this top tier (e.g., DeepSeek-V3.1 with tools) demonstrate a qualitative leap in capability. They have transcended the reasoning bottleneck and exhibit robust, multi-step planning and synthesis. Their behavior is analogous to the More Knowledgeable Other, seamlessly integrating tool-based exploration into a coherent reasoning process to solve problems far beyond their intrinsic reach. Achieving this level of performance signifies the emergence of a truly capable agent that can autonomously navigate complex problem spaces.

In summary, the ZPD Exam serves not merely as a leaderboard but as a powerful diagnostic instrument. Its tiered results provide a fine-grained analysis of an agent's developmental stage—from what it knows (intrinsic), to what it can learn to do with support (ZPD), to what it has mastered. This allows us to pinpoint critical reasoning faculties that require improvement, thereby charting a clear path toward more autonomous and capable AI agents.

# 4 EXPERIMENTS

## 4.1 EXPERIMENTAL SETUP

**Training Data Synthesis.** We synthesize training trajectories using a tool-augmented DeepSeek-V3.1 (Liu et al., 2024), following the iterative tool-calling and summarization paradigm from WebResearcher (Qiao et al., 2025; Chen et al., 2025). Each trajectory is generated through a multi-round process adhering to the ReAct (Yao et al., 2023), comprising a sequence of round-wise reasoning reports and observations after the corresponding tool calls. In each round, the model generates a reasoning report that summarizes accumulated evidence, analyzes progress towards the research question, and specifies the next action—either invoking a new tool or outputting a final answer.

**Baselines and Fine-Tuning.** We compare our proposed AgentFrontier dataset against three prominent, multi-disciplinary agent-tuning datasets: TaskCraft (Shi et al., 2025), MegaScience (Fan et al., 2025) and MiroVerse (MiroMind-Data-Team, 2025). To ensure a fair comparison, we first curate a high-quality subset of 12,000 trajectories for each dataset via rejection sampling, retaining only instances where the final answer is perfectly correct. As shown in Table 1, our AgentFrontier dataset exhibits a more balanced and diverse tool-use distribution compared to the baselines, with substantial usage across scholar, browser, and code tools. This reflects its focus on complex reasoning problem-solving. For rejection sampling fine-tuning (RFT), we normalize the training data volume to 25,600 rounds for each dataset, with each round capped at 40,960 tokens, and train for 3 epochs.

Table 1: Statistics of trajectories across the training datasets. Avg. Rounds and Avg. Calls are computed per trajectory.

| Dataset | Rounds | Avg. Tool Calls | | | |
|---|---|---|---|---|---|
| | | Search | Scholar | Browser | Code |
| TaskCraft | 3.38 | 1.04 | 0.14 | 1.19 | 0.01 |
| MegaScience | 2.68 | 0.26 | 0.56 | 0.49 | 0.37 |
| MiroVerse | 2.18 | 0.12 | 0.04 | 0.09 | 0.93 |
| AgentFrontier | 3.32 | 0.32 | 0.66 | 0.82 | 0.52 |

**Models and Evaluation.** We conduct experiments on the Qwen3 model family (Yang et al., 2025), including both dense (Qwen3-8B, Qwen3-32B) and Mixture-of-Experts (Qwen3-30B-A3B-Thinking-2507) variants. We evaluate performance on four challenging benchmarks designed to probe high-level reasoning across diverse disciplines: HLE (Phan et al., 2025), ZPD Exam, R-Bench (Guo et al., 2025b) and xBench-ScienceQA (Xbench-Team, 2025). For evaluating the correctness of final answers, we employ an **LLM-as-a-Judge**. Specifically, we use o3-mini (OpenAI, 2025b) as the judge, guided by the official strict evaluation prompt from HLE (Phan et al., 2025). All model generations use nucleus sampling with a temperature of 0.6 and a top-p of 0.95.

## 4.2 MAIN RESULTS

**Overall Performance Across Benchmarks.** Table 2 illustrates the performance of the Qwen3-series models after fine-tuning. Models trained on AgentFrontier consistently achieve state-of-the-art results, decisively outperforming all baseline datasets across all four benchmarks. In contrast, the performance of competing datasets such as TaskCraft (Shi et al., 2025), MegaScience (Fan et al., 2025), and MiroVerse (MiroMind-Data-Team, 2025) is inconsistent; while each may show strength on a particular benchmark, none demonstrates the robust, cross-domain superiority imparted by AgentFrontier. This trend of superior and consistent performance holds across other model backbones as well.

**Subject-Level Dominance on the HLE Benchmark.** To investigate the source of this performance advantage, we conduct a fine-grained analysis on the particularly demanding Humanity's Last Exam (HLE) (Phan et al., 2025) benchmark, examining results across eight academic disciplines with various model backbones (Table 3). For both the Qwen3-8B and Qwen3-32B backbones, models trained on AgentFrontier exhibit remarkable breadth, securing the top performance in six and seven out of the eight subjects, respectively. This subject-level dominance translates to a significant lead in overall average scores, with AgentFrontier surpassing the next-best dataset by 3.8 and 3.9 absolute points on the 8B and 32B models, respectively. The advantage becomes even more pronounced with the Qwen3-30B-A3B model, where fine-tuning on AgentFrontier outperforms all competing datasets in every single subject. This comprehensive superiority results in a final average score of 25.67%, rep-

Table 2: Performance comparison on four multi-disciplinary benchmarks. Scores are reported as "mean ± confidence interval". The **best** score is highlighted, and the second-best is underlined.

| RFT Dataset | Tools | Evaluation on Four Multi-disciplinary Benchmarks | | | |
|---|---|---|---|---|---|
| | | HLE (text-only) | ZPD Exam-v1 | RBench-T | xBench-SciQA |
| *Backbone: Qwen3-8B* | | | | | |
| $-_{(no-finetuning)}$ | ✗ | $4.0_{\pm0.76}$ | $5.3_{\pm0.58}$ | $55.0_{\pm1.19}$ | $20.0_{\pm3.48}$ |
| $-_{(no-finetuning)}$ | ✓ | $5.9_{\pm0.84}$ | $35.2_{\pm1.23}$ | $58.2_{\pm1.23}$ | $24.0_{\pm3.27}$ |
| TaskCraft | ✓ | $14.6_{\pm1.26}$ | $\mathbf{87.5}_{\pm0.85}$ | $64.3_{\pm1.19}$ | $30.0_{\pm3.72}$ |
| MegaScience | ✓ | $14.2_{\pm1.26}$ | $84.7_{\pm0.93}$ | $62.3_{\pm1.20}$ | $36.0_{\pm3.90}$ |
| MiroVerse | ✓ | $15.0_{\pm1.28}$ | $84.5_{\pm0.93}$ | $62.8_{\pm1.20}$ | $32.0_{\pm3.79}$ |
| **AgentFrontier** | ✓ | $\mathbf{18.8}_{\pm1.32}$ | $86.8_{\pm0.92}$ | $\mathbf{67.2}_{\pm1.18}$ | $\mathbf{40.0}_{\pm3.85}$ |
| *Backbone: Qwen3-32B* | | | | | |
| $-_{(no-finetuning)}$ | ✗ | $7.3_{\pm0.98}$ | $5.8_{\pm0.60}$ | $60.9_{\pm1.15}$ | $37.0_{\pm3.96}$ |
| $-_{(no-finetuning)}$ | ✓ | $8.4_{\pm0.92}$ | $48.6_{\pm1.28}$ | $65.1_{\pm1.18}$ | $39.0_{\pm3.92}$ |
| TaskCraft | ✓ | $18.4_{\pm1.38}$ | $\mathbf{91.1}_{\pm0.73}$ | $66.2_{\pm1.18}$ | $40.0_{\pm3.98}$ |
| MegaScience | ✓ | $18.5_{\pm1.38}$ | $89.6_{\pm0.78}$ | $68.4_{\pm1.16}$ | $40.0_{\pm3.98}$ |
| MiroVerse | ✓ | $19.9_{\pm1.42}$ | $87.7_{\pm0.84}$ | $67.4_{\pm1.16}$ | $43.0_{\pm4.02}$ |
| **AgentFrontier** | ✓ | $\mathbf{23.8}_{\pm1.52}$ | $90.9_{\pm0.73}$ | $\mathbf{70.3}_{\pm1.14}$ | $\mathbf{51.0}_{\pm4.06}$ |
| *Backbone: Qwen3-30B-A3B-Thinking-2507* | | | | | |
| $-_{(no-finetuning)}$ | ✗ | $9.2_{\pm1.06}$ | $4.9_{\pm0.56}$ | $51.2_{\pm1.07}$ | $32.0_{\pm3.79}$ |
| $-_{(no-finetuning)}$ | ✓ | $10.2_{\pm1.08}$ | $47.2_{\pm1.28}$ | $55.1_{\pm1.13}$ | $40.0_{\pm3.98}$ |
| TaskCraft | ✓ | $19.9_{\pm1.42}$ | $90.1_{\pm0.76}$ | $72.3_{\pm1.11}$ | $44.0_{\pm4.08}$ |
| MegaScience | ✓ | $20.2_{\pm1.42}$ | $90.0_{\pm0.77}$ | $73.1_{\pm1.10}$ | $48.0_{\pm4.08}$ |
| MiroVerse | ✓ | $19.6_{\pm1.42}$ | $86.7_{\pm0.87}$ | $70.6_{\pm1.13}$ | $49.0_{\pm4.08}$ |
| **AgentFrontier** | ✓ | $\mathbf{25.7}_{\pm1.50}$ | $\mathbf{91.4}_{\pm0.79}$ | $\mathbf{74.4}_{\pm1.13}$ | $\mathbf{54.0}_{\pm4.01}$ |

resenting a 178% and 152% relative improvement over the original base model in settings without and with tool augmentation, respectively. These results indicate that as model capacity increases, the rich, multi-step reasoning trajectories within AgentFrontier become increasingly effective at unlocking expert-level problem-solving capabilities across a wide spectrum of academic fields.

## 5 ANALYSIS

### 5.1 SENSITIVITY TO LKP / MKO CONFIGURATION

We conduct an ablation study on the Less Knowledgeable Peer (LKP) and More Knowledgeable Other (MKO) configurations to assess our framework's sensitivity. The study investigates the trade-off between **synthesis efficiency** (data yield) and **data complexity** (reasoning depth), aiming to validate that our chosen configuration strikes an effective balance.

**Experimental Setup.** We evaluate three LKP/MKO configurations on a 1,000-sample subset of $D_{\text{refined}}$ to probe varying capability gaps. Note that DeepSeek-V3.1 possesses stronger reasoning and agentic abilities than DeepSeek-R1 (Liu et al., 2024; Guo et al., 2025a). The configurations are: **(1) Original (Balanced Gap)**, using DeepSeek-R1 (no tools) as LKP and DeepSeek-V3.1 (with tools) as MKO; **(2) Wider Gap**, replacing the LKP with a weaker Qwen3-30B-A3B; and **(3) Narrower Gap**, using DeepSeek-R1 for both roles, where the MKO is distinguished only by its access to tools.

**Results and Analysis.** As presented in Table 4, a **wider gap** (Config. 2) with a weaker LKP increases the data yield by 44.1% but at the cost of complexity, evidenced by a sharp decrease in average rounds (↓44.3%) and tool calls (↓63.4%). This results in simpler data, less effective for advancing model capabilities. Conversely, a **narrower gap** (Config. 3), where models differ only in tool access, maintains high complexity but suffers a 27.5% drop in yield, rendering it inefficient for large-scale synthesis. These results empirically validate that our chosen **original configuration**

Table 3: Accuracy on the Humanity's Last Exam (full text-only set). Results are reported across major knowledge domains. Each block corresponds to a different Qwen3 backbone. Numbers with a colored background denote the best within each block; underlined numbers denote the second best.

| RFT Dataset | Tools | Domain Accuracy on Humanity's Last Exam (%) | | | | | | | | |
|---|---|---|---|---|---|---|---|---|---|---|
| | | Math | CS/AI | Bio./Med. | Physics | Humanities | Chem. | Eng. | Other | Avg. |
| *Backbone: Qwen3-8B* | | | | | | | | | | |
| $-_{(no-finetuning)}$ | ✗ | 6.46 | 2.65 | 5.88 | 0.99 | 3.63 | 1.00 | 6.45 | 1.61 | 4.00 |
| $-_{(no-finetuning)}$ | ✓ | 6.26 | 3.54 | 9.05 | 2.48 | 7.25 | 7.00 | 6.45 | 5.14 | 5.94 |
| TaskCraft | ✓ | 16.21 | 10.62 | 14.93 | 6.44 | 22.80 | 9.00 | 9.68 | 15.43 | 14.58 |
| MegaScience | ✓ | 14.56 | 10.62 | **18.10** | 5.94 | 21.76 | 9.00 | **12.90** | 16.57 | 14.21 |
| MiroVerse | ✓ | 17.33 | 10.62 | 15.38 | 5.94 | 21.24 | 8.00 | 6.45 | 17.71 | 15.00 |
| **AgentFrontier** | ✓ | **22.46** | **14.16** | 16.74 | **10.40** | **24.35** | **11.00** | 6.45 | **19.43** | **18.80** |
| *Backbone: Qwen3-32B* | | | | | | | | | | |
| $-_{(no-finetuning)}$ | ✗ | 8.72 | 5.75 | 10.41 | 0.50 | 7.77 | 8.00 | 6.45 | 5.14 | 7.34 |
| $-_{(no-finetuning)}$ | ✓ | 10.97 | 5.31 | 9.05 | 4.95 | 7.25 | 5.00 | 6.45 | 4.57 | 8.36 |
| TaskCraft | ✓ | 20.72 | 14.16 | 16.74 | 8.91 | 25.39 | 14.00 | 14.52 | 20.57 | 18.43 |
| MegaScience | ✓ | 21.23 | 14.60 | 14.93 | 6.44 | 29.02 | 12.00 | 11.29 | 21.71 | 18.52 |
| MiroVerse | ✓ | 22.56 | 14.16 | 16.74 | 10.40 | **34.72** | 12.00 | 6.45 | 20.57 | 19.92 |
| **AgentFrontier** | ✓ | **28.21** | **16.81** | **18.10** | **15.84** | 30.57 | **15.00** | **19.35** | **24.00** | **23.82** |
| *Backbone: Qwen3-30B-A3B-Thinking-2507* | | | | | | | | | | |
| $-_{(no-finetuning)}$ | ✗ | 13.03 | 7.96 | 8.14 | 3.47 | 7.25 | 5.00 | 8.06 | 2.86 | 9.24 |
| $-_{(no-finetuning)}$ | ✓ | 13.13 | 7.96 | 6.33 | 1.98 | 11.92 | 10.00 | 6.45 | 10.29 | 10.17 |
| TaskCraft | ✓ | 24.62 | 12.39 | 16.29 | 7.92 | 21.76 | 19.00 | 12.90 | 22.29 | 19.87 |
| MegaScience | ✓ | 23.69 | 14.60 | 20.81 | 9.90 | 26.94 | 15.00 | 8.06 | 18.29 | 20.15 |
| MiroVerse | ✓ | 23.38 | 12.39 | 20.81 | 9.41 | 24.87 | 7.00 | 11.29 | 22.86 | 19.64 |
| **AgentFrontier** | ✓ | **29.85** | **16.81** | **21.27** | **17.82** | **31.61** | **22.00** | **14.52** | **28.00** | **25.67** |

achieves a crucial trade-off, ensuring both scalable data generation and sufficient data complexity. This demonstrates our model selection is a deliberate strategy to optimize this balance, not an arbitrary choice.

Table 4: Ablation study on LKP/MKO configurations, analyzing the trade-off between ZPD data yield and data complexity. The ZPD Data Yield is defined as the number of valid $D_{ZPD}$ samples divided by the total candidate samples. Our original configuration (in bold) demonstrates a superior balance. S/Sc/B/C denotes Search, Scholar, Browser, and Code tools respectively.

| Configuration (LKP / MKO) | ZPD Data Yield (%) | Avg. Rounds | Avg. Tool Calls | Tool Usage Dist. (S/Sc/B/C) |
|---|---|---|---|---|
| **1. DS-R1 / DS-V3.1+T (Original)** | **33.1** | **3.32** | **2.32** | 0.32 / 0.66 / 0.82 / 0.52 |
| 2. Qwen3-30B / DS-V3.1+T (Wider Gap) | 47.7 (↑44.1%) | 1.85 (↓44.3%) | 0.85 (↓63.4%) | 0.18 / 0.23 / 0.36 / 0.08 (all ↓) |
| 3. DS-R1 / DS-R1+T (Narrower Gap) | 24.0 (↓27.5%) | 2.99 (≈) | 1.99 (≈) | 0.19 / 0.67 / 0.58 / 0.55 |

## 5.2 BoN Analysis: Validating Difficulty Richness & Potential for RL Training

To assess the difficulty distribution of AgentFrontier and the latent capabilities of the RFT model, we conducted a Best-of-N (BoN) analysis. On a held-out validation set of 300 samples, we generated $N = 8$ independent solution trajectories for each task and measured the success rate if at least one of the $N$ attempts was correct (pass@$N$).

As shown in Figure 4, the accuracy dramatically increases from 21.7% at pass@1 to 40.7% at pass@8. This 19.0-point improvement provides two key insights. **First, it validates the designed difficulty of AgentFrontier:** the dataset is not a binary mix of trivial and impossible tasks. Instead, it presents a challenging frontier where initial attempts may fail, but success is achievable through exploration. This provides a rich learning signal beyond superficial pattern matching. **Second, it highlights the significant potential for subsequent reinforcement learning (RL)** While supervised

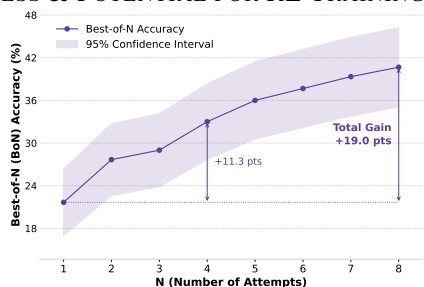

Figure 4: Best-of-N accuracy of our RFT Qwen3-30B-A3B on a 300-sample validation set from AgentFrontier.

fine-tuning (SFT) trains the model on a single reference solution, the large gap between pass@1 and pass@8 confirms that for problems the model fails to solve on the first attempt, its policy distribution contains diverse and successful alternative trajectories. This is a crucial precondition for effective RL, ensuring that exploration can discover high-reward experiences necessary for effective policy optimization. Therefore, AgentFrontier serves not only as a robust training resources for SFT but also as a strong foundation for RL to further unlock an agent's problem-solving potential.

## 5.3 Why AgentFrontier Excels: Dissecting the Leap in Reasoning & Tool-Use

**From Shallow Retrieval to Deep Causal Reasoning.** Figure 5 reveals the performance dynamics that underscore AgentFrontier's superiority. The vast majority (95%) of problems are solved within a 15-round horizon, a critical window in which our RFT dataset consistently outperforms all fine-tuning dataset baselines. This advantage is a principled consequence of our data generation strategy rooted in the Zone of Proximal Development. By curating tasks that are unsolvable by the base model yet solvable with external scaffolding, we create training instances of optimal difficulty. This forces the model to abandon simplistic, single-source retrieval and instead master knowledge fusion—the non-trivial meta-skill of integrating disparate information streams into a coherent solution. The agent learns not merely what information to retrieve, but how to synthesize it, shifting from shallow pattern-matching to in-depth causal reasoning.

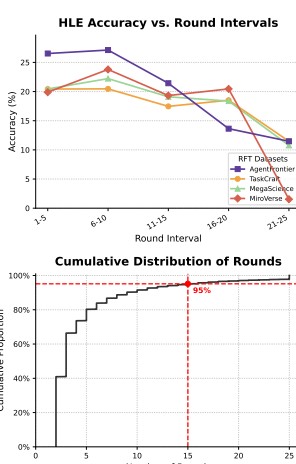

Figure 5: Accuracy vs. number of rounds on 4 datasets.

**From High-Volume Invocation to High-Efficacy Orchestration.** The design philosophy of AgentFrontier prioritizes the cultivation of strategic tool orchestrators over rote tool callers. Unlike datasets that promote skewed tool dependencies (e.g., code-centric MiroVerse or search-centric TaskCraft), AgentFrontier promotes a balanced tool-use distribution (Table 1). This forces the agent to develop a sophisticated understanding of inter-tool synergy rather than mastering a single tool in isolation. The results on the HLE benchmark (Table 5) confirm this empirical payoff. Our agent achieves a macro-average conditional tool accuracy of 26.3%—a significant leap from the 21% plateau of competitors—with a comparable number of interactions. This demonstrates that agent capability stems not from the volume of tool calls, but their efficacy. Our method trains the model to transition from high-volume, low-yield tool usage to precise, high-efficacy orchestration, which is a crucial step toward creating more resourceful agents.

Table 5: Tool usage statistics for the Qwen3-30B-A3B agent on the HLE text-only test set (2154 problems). Each column block shows performance after RFT on a different dataset. We report average usage per round and conditional tool accuracy (Acc, %), defined as the success rate for tasks that use the tool. The final row details overall metrics. Best results are in **bold**.

| Tool / Metric | TaskCraft | | MegaScience | | MiroVerse | | AgentFrontier | |
|---|---|---|---|---|---|---|---|---|
| | Usage | Acc (%) | Usage | Acc (%) | Usage | Acc (%) | Usage | Acc (%) |
| Search | 0.68 | 19.6 | 0.67 | 20.3 | **0.73** | 20.4 | **0.73** | **24.9** |
| Scholar | 0.78 | 21.0 | **0.98** | 20.3 | 0.87 | 20.6 | 0.89 | **25.4** |
| Browser | 1.24 | 25.2 | 1.39 | 23.4 | **1.47** | 22.7 | 1.32 | **29.8** |
| Code | 0.52 | 18.1 | 0.65 | 18.6 | **0.67** | 18.4 | 0.63 | **24.9** |
| **Overall** (Rounds/Acc.) | 4.21 | 21.0 | 4.70 | 20.6 | **4.74** | 20.5 | 4.57 | **26.3** |

## 5.4 Holistic Agentic Training

**Setup.** We further investigate the benefits of a holistic training pipeline that incorporates continued pre-training (CPT) and post-training. Due to the large-scale GPU computation in CPT, this study is conducted only on Qwen3-30B-A3B-Thinking-2507 and our AgentFrontier data. The holistic training pipeline consists of two stages: (1) **Continued Pre-training (CPT)**: One epoch over 50B tokens, comprising 1 million summarized text chunks and 20 million knowledge-intensive QA pairs. (2) **Rejection Sampling Fine-tuning (RFT)**: Three epochs on 12,000 high-quality trajectories.

**Evaluation.** We conducted a comprehensive evaluation of our model, AgentFrontier-30B-A3B, against a broad spectrum of competitors: proprietary LLMs (OpenAI, 2024; anthropic, 2025; Deep-Mind, 2025) with tools, proprietary and prominent open-source deep-research agents (OpenAI, 2025a; Google, 2025; MoonshotAI, 2025; Wu et al., 2025; Li et al., 2025a; Tao et al., 2025).

**Results.** As shown in Table 6, our holistically trained agent not only sets a new state-of-the-art among open-source models but also competes effectively with significantly larger, proprietary agents. The final row isolates the contribution of CPT, which consistently boosts performance across all benchmarks (+2.9 on HLE, +7.0 on xBench-ScienceQA). Notably, CPT yields a +2.0 point gain on ZPD Exam, where the RFT-only model's performance was already near-saturation. This provides strong evidence that strengthening a model's foundational knowledge via CPT directly enhances its capacity for complex agentic tasks.

Table 6: AgentFrontier-30B outperforms SOTA agents on four multi-disciplinary benchmarks. The performance gain from our CPT is shown in the final row. † marks results from official reports.

| Agents | HLE | ZPD Exam | RBench-T | xBench-SciQA |
|---|---|---|---|---|
| *Proprietary LLMs with Tools & Deep-Research Agents* | | | | |
| GPT-4o | 4.8 | 51.3 | 48.5 | 15.0 |
| Claude 4 Sonnet | 14.3 | 86.6 | 71.1 | 47.0 |
| Gemini 2.5 Flash | 12.6 | 58.1 | 75.8 | 39.0 |
| OpenAI DeepResearch | 26.6† | – | – | – |
| Gemini DeepResearch | 26.9† | – | – | – |
| Kimi-Researcher | 26.9† | – | – | – |
| *Open-source Agents* | | | | |
| WebDancer-QwQ-32B | 6.4 | 51.8 | 67.6 | 38.0 |
| WebSailor-72B | 9.2 | 62.1 | 44.9 | 27.0 |
| WebShaper-72B | 8.0 | 54.4 | 66.8 | 29.0 |
| *AgentFrontier-30B-A3B (Ours)* | | | | |
| **RFT only** | 25.7 | 91.4 | 74.4 | 54.0 |
| **CPT+RFT** | **28.6** | **93.4** | **77.1** | **61.0** |
| Δ **(CPT gain)** | **+2.9** | **+2.0** | **+2.7** | **+7.0** |

## 5.5 CASE STUDY

A qualitative analysis on an HLE case (Phan et al., 2025) (Appendix E) further illustrates our agent's reasoning process. In a complex clinical scenario, OpenAI DeepResearch (OpenAI, 2025a) agent exhibited **diagnostic fixation**, misdiagnosing *Charcot Arthropathy* by focusing on common negative findings like sterile synovial fluid. In contrast, our AgentFrontier agent correctly identified the key anomaly: the patient's paradoxical worsening on prednisone. It hypothesized that this was due to a latent infection unmasked by immunosuppression, rather than an inflammatory rebound. This triggered a targeted inquiry, using a literature search to confirm that *Chronic Osteomyelitis* can present with sterile aspirates and is exacerbated by steroids. This progression from identifying an anomaly to forming a hypothesis and validating it with targeted research demonstrates AgentFrontier's advanced research capabilities.

## 6 RELATED WORK

**Data Synthesis for LLM Agents.** Synthesizing high-quality data is critical for advancing LLM agents that require complex reasoning and tool use (Zeng et al., 2025; Liu et al., 2025; Zhou et al., 2024). Initial efforts replaced costly manual curation with programmatic generation, creating agentic tasks with verifiable solution trajectories (Shi et al., 2025; Hongjin et al., 2025; Huang et al., 2025). Subsequent research aimed to enhance data quality by grounding synthesis in external knowledge sources like scientific documents (Fan et al., 2025; Feng et al., 2025). While these approaches increase factual richness, they often produce tasks solvable via localized information retrieval, rather than promoting the deep knowledge integration essential for complex research (OpenAI, 2025a). A central challenge remains the precise calibration of task difficulty. Without a principled control mechanism, synthetic data risks being too simple for effective learning or too complex to yield a usable training signal (Li et al., 2025b). These strategies rely on heuristics like incremental constraint addition (Patel et al., 2025) or probes to distinguish reasoning from recitation (Yan et al., 2025), yet lack a principled framework to calibrate difficulty for scaffolding complex reasoning.

## 7 CONCLUSION

In this work, we presented a novel data synthesis paradigm based on the Zone of Proximal Development (ZPD) theory. Our framework co-generates a targeted training resources and a self-evolving ZPD Exam to progressively enhance and evaluate agentic reasoning. The resulting model, AgentFrontier-30B-A3B, validates our approach by achieving state-of-the-art results on challenging expert-level multi-disciplinary benchmarks, surpassing even significantly larger proprietary agents. This work demonstrates that a principled, pedagogical approach to data synthesis is a highly effective, if not essential, strategy for cultivating advanced reasoning abilities in a data-efficient manner.

## ACKNOWLEDGMENTS

This study was supported by grants from the National Science Foundation of China (T2541010), the National Key R&D Program of China (2024YFF1207100, 2024YFF1207103), and Beijing National Research Center for Information Science and Technology (BNRist). The funders had no roles in study design, data collection and analysis, publication decisions, or manuscript preparation.

## ETHICS STATEMENT

All authors of this work have read and agree to adhere to the ICLR Code of Ethics. The corpora used for data synthesis in our research are sourced from publicly available documents. We have ensured that our use of this data complies with all applicable terms of use and licenses provided by the data owners.

## REPRODUCIBILITY STATEMENT

For immediate verification during the review process, the paper and its supplementary materials already include:

- **Evaluation Data:** The complete question-and-answer sets for all evaluation benchmarks used in our study.
- **Generated Examples:** A curated set of 100 question-and-answer examples from our AgentFrontier pipeline, provided for qualitative analysis.
- **Novel Benchmark:** The full ZPD Exam-v1 benchmark proposed in this work.
- **Implementation Details:** Detailed descriptions of training hyperparameters, tool implementation, and evaluation setup are available in Appendix C and D.
- **Prompts:** The exact prompts employed for our LLM-as-a-Judge, data filtering mechanisms and iterative agentic refinement are provided in Appendix F.

Together, these resources offer a transparent and direct pathway for verifying our findings and serve as a foundation for future research.

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

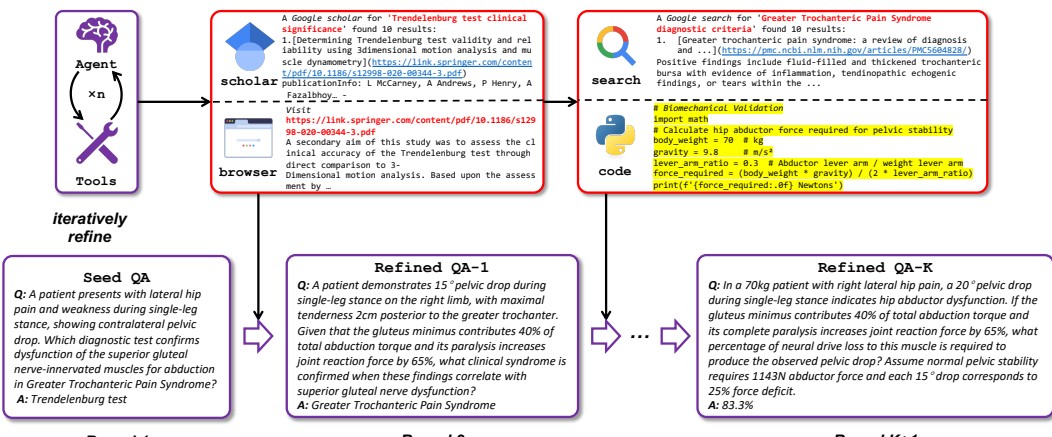

Figure 6: An overview of our iterative refinement process. We start with a biomedical seed QA, which is then refined into a complex diagnostic reasoning problem by synthesizing knowledge from academic literature. Finally, this problem is evolved into a practical computational challenge grounded in a real-world application, a process involving web search and programmatic validation.

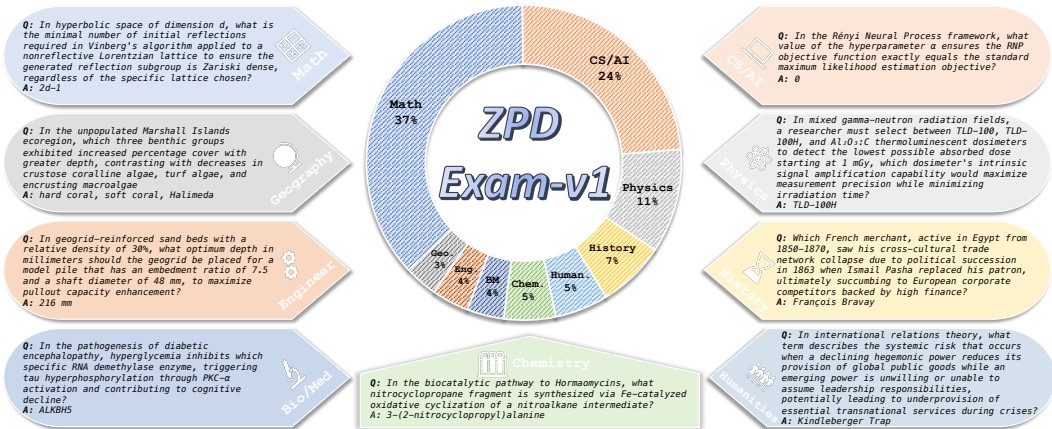

Figure 7: The ZPD Exam-v1 consists of 1024 questions categorized into 9 disciplines: Mathematics, Computer Science / Artificial Intelligence, Physics, History, Humanities, Chemistry, Biology / Medicine, Engineering, and Geography.

# APPENDIX

## A  THE USE OF LARGE LANGUAGE MODELS (LLMs)

During the preparation of this manuscript, we utilized Large Language Models (LLMs) for assistance with language proofreading (including grammar, spelling, and word choice) and for generating LaTeX code for tables and figures. The core intellectual contributions, including research ideation, analysis, and the substantive writing, are entirely the work of the authors.

## B  MORE RELATED WORK

**Multi-disciplinary Benchmark.** The evaluation of advanced reasoning in large language models (LLMs) was pioneered by MMLU (Hendrycks et al., 2021), which set the standard for assessing multi-disciplinary knowledge. This led to a wave of subsequent benchmarks (Rein et al., 2023; Wang et al., 2024; Du et al., 2025; Guo et al., 2025b; Xbench-Team, 2025) targeting undergraduate or graduate level knowledge. However, the rapid progress of frontier models (OpenAI, 2025b;

DeepMind, 2025; anthropic, 2025) is causing performance saturation on these static benchmarks, reducing their effectiveness in differentiating top-tier models. While newer benchmarks like Humanity's Last Exam (Phan et al., 2025) increase difficulty through expert curation, they remain fixed assessments. In contrast, our work introduces the ZPD Exam, a self-evolving evaluation framework that adapts in lockstep with model capabilities, providing a consistently challenging frontier for LLM agent evaluation.

**Deep-Research Agents.** Deep-research agent, a system built upon large reasoning models (LRMs), is designed to automate multi-step search and reasoning. It empowers users to complete complex, cross-domain information synthesis and in-depth research tasks in minutes, a process that would otherwise require hours of human effort. Proprietary agents (OpenAI, 2025a; Google, 2025; Anthropic, 2025; xAI, 2025; Perplexity, 2025; MoonshotAI, 2025) have demonstrated impressive capabilities in complex, multi-step research tasks. The open-source community has fostered a rich ecosystem of transparent and reproducible agents (Jin et al., 2025; Li et al., 2025c;d; Tao et al., 2025; Li et al., 2025a; Qiao et al., 2025). These efforts typically leverage explicit planning, tool-use, and web navigation to emulate human research processes, advancing the field through shared methodologies.

## C DATA ENGINE DETAILS

This section provides a detailed breakdown of the hyperparameters, procedural logic, and computational costs associated with the AgentFrontier Data Engine, as outlined in Algorithm 1. These details are provided to ensure the transparency and reproducibility of our data synthesis framework.

### C.1 HYPERPARAMETER CONFIGURATION

The data generation pipeline is governed by several key hyperparameters that control the granularity of data sourcing, the complexity of generated questions, and the strictness of the filtering process. Our configuration is as follows:

- **Thematic Coherence Threshold ($\tau_{\mathbf{theme}}$):** Set to **0.8**. This value determines the minimum semantic similarity required between text chunks to form a "composite unit" for seed question generation. A higher value ensures that initial questions are synthesized from thematically tighter content, promoting knowledge fusion.

- **Nearest Neighbors for Seeding ($k_{\mathbf{nn}}$):** Set to **10**. During seed generation, for each text chunk, we retrieve its $k_{\mathrm{nn}}$ nearest neighbors to search for coherent triplets. This balances computational efficiency with a sufficiently large search space for discovering novel combinations.

- **Maximum Refinement Iterations ($K_{\mathbf{max}}$):** Set to **30**. This parameter defines the maximum number of complexity escalation steps for any given QA pair in Stage II. This upper bound prevents infinite loops and manages computational resources.

- **Best-of-N (BoN) Verification Size ($N$):** Set to **3**. In the ZPD-filtering stage, the More Knowledgeable Other ($\mathcal{A}_{\mathrm{MKO}}$) makes $N$ independent attempts to solve a problem. This helps to reduce the variance in the agent's performance and provides a more reliable signal of whether a task is solvable.

- **Diversity Filter Threshold ($\epsilon$):** Set to **0.7**. To ensure dataset diversity, a new QA pair is discarded if its question's semantic similarity to any existing question in $\mathcal{D}_{\mathrm{ZPD}}$ exceeds this threshold. The similarity is measured by a state-of-the-art reranker model.

### C.2 AGENTIC REFINEMENT AND STOPPING CRITERION

The core of our data engine is the iterative refinement loop (Stage II), driven by the agent $\mathcal{A}_{\mathrm{refine}}$. The goal of the escalation operator, $\Psi_{\mathrm{escalate}}$, is to progressively increase the cognitive load required to answer a question. This is achieved by prompting the agent to perform a series of enrichment actions, including but not limited to: expanding the question with new, relevant concepts discovered through tool use; abstracting a general principle from specific examples; grounding the problem in a more complex, realistic context; or transforming a qualitative problem into a quantitative one requiring computation.

---

**Algorithm 1** AgentFrontier Data Engine Pipeline

---

   **Input:**
      $\mathcal{C}_{\text{raw}}$: Raw document corpus
      $\Phi_{\text{chunk}}$: Chunking model
      $\mathcal{M}_{\text{gen}}, \mathcal{A}_{\text{refine}}, \mathcal{A}_{\text{LKP}}, \mathcal{A}_{\text{MKO}}$: Models and agents
      Sim, IsCorrect, IsSolvableBy: Similarity and evaluation functions
      $\tau_{\text{theme}}, K, N, \epsilon, k_{\text{nn}}$: Hyperparameters (thematic threshold, escalation steps, BoN size,
          redundancy threshold, number of neighbors)
   **Output:**
      $\mathcal{D}_{\text{ZPD}}$: Calibrated training dataset for post-training
      $\mathcal{D}_{\text{pretrain}}$: Dataset for continued pre-training
      $\mathcal{D}_{\text{human}}$: Dataset for human review

1:  **procedure** GENERATEZPDDATA($\mathcal{C}_{\text{raw}}, \dots$)
2:     $\mathcal{D}_{\text{ZPD}}, \mathcal{D}_{\text{pretrain}}, \mathcal{D}_{\text{human}} \leftarrow \emptyset, \emptyset, \emptyset$
                                           ▷ **Stage I: Seed Question Generation**
3:     $\mathcal{C}_{\text{chunk}} \leftarrow \bigcup_{d \in \mathcal{C}_{\text{raw}}} \Phi_{\text{chunk}}(d)$          ▷ Preprocess corpus into semantic chunks
4:     $\mathcal{V}_{\text{index}} \leftarrow \text{BuildVectorIndex}(\mathcal{C}_{\text{chunk}})$          ▷ Build index for efficient search
5:     $\mathcal{D}_{\text{seed}} \leftarrow \emptyset$
6:     **for** each chunk $c_i \in \mathcal{C}_{\text{chunk}}$ **do**
7:         $\mathcal{N}_i \leftarrow \text{FindNearestNeighbors}(c_i, \mathcal{V}_{\text{index}}, k_{\text{nn}})$   ▷ Find k-NN for efficient combination
8:         **for** each pair $(c_j, c_k)$ from $\mathcal{N}_i$ **do**
9:             **if** $\text{Sim}(c_i, c_j) > \tau_{\text{theme}} \wedge \text{Sim}(c_i, c_k) > \tau_{\text{theme}} \wedge \text{Sim}(c_j, c_k) > \tau_{\text{theme}}$ **then**
10:               $(q_0, a_0) \leftarrow \mathcal{M}_{\text{gen}}(\{c_i, c_j, c_k\})$       ▷ Generate QA from thematic unit
11:               $\mathcal{D}_{\text{seed}} \leftarrow \mathcal{D}_{\text{seed}} \cup \{(q_0, a_0)\}$
12:            **end if**
13:         **end for**
14:     **end for**
                             ▷ **Stages II & III: Iterative Escalation and ZPD Calibration**
15:     $\mathcal{V}_{\text{ZPD}} \leftarrow \text{BuildVectorIndex}(\emptyset)$         ▷ Initialize index for ZPD-set diversity check
16:     **for** each $(q_0, a_0)$ in $\mathcal{D}_{\text{seed}}$ **do**
17:         $(q, a) \leftarrow (q_0, a_0)$
                                   ▷ **Stage II: Agentic Refinement**
18:         **for** $k = 1$ to $K$ **do**              ▷ Iteratively escalate complexity
19:             $(q, a) \leftarrow \Psi_{\text{escalate}}(q, a, \mathcal{A}_{\text{refine}})$     ▷ e.g., Expand, Abstract, Ground, etc.
20:         **end for**
                                    ▷ **Stage III: ZPD-based Filtering**
21:         **if** IsSolvableBy($\mathcal{A}_{\text{LKP}}, q, a$) **then**     ▷ Check if too easy for Less Knowledgeable Peer
22:             $\mathcal{D}_{\text{pretrain}} \leftarrow \mathcal{D}_{\text{pretrain}} \cup \{(q, a)\}$
23:         **else**                      ▷ Challenging for LKP, now verify with MKO
24:             $S_{\text{solutions}} \leftarrow \{\mathcal{A}_{\text{MKO}}(q) \text{ for } i = 1 \dots N\}$ ▷ Best-of-N by More Knowledgeable Other
25:             **if** $\exists s \in S_{\text{solutions}}$ s.t. IsCorrect($s, a$) **then**    ▷ Verified as solvable, thus within ZPD
26:                $q_{\text{nearest}} \leftarrow \text{FindNearestNeighbor}(q, \mathcal{V}_{\text{ZPD}})$
27:                **if** $q_{\text{nearest}} = \emptyset$ or $\text{Sim}(q, q_{\text{nearest}}) < \epsilon$ **then**       ▷ Filter for diversity
28:                   $\mathcal{D}_{\text{ZPD}} \leftarrow \mathcal{D}_{\text{ZPD}} \cup \{(q, a)\}$
29:                   $\text{UpdateVectorIndex}(\mathcal{V}_{\text{ZPD}}, q)$
30:                **end if**
31:             **else**                 ▷ Unsolvable by MKO, potentially flawed or too hard
32:                $\mathcal{D}_{\text{human}} \leftarrow \mathcal{D}_{\text{human}} \cup \{(q, a)\}$
33:             **end if**
34:         **end if**
35:     **end for**
36:     **return** $\mathcal{D}_{\text{ZPD}}, \mathcal{D}_{\text{pretrain}}, \mathcal{D}_{\text{human}}$
37: **end procedure**

---

The iterative escalation is guided by a principled stopping criterion tied to the ZPD framework: for a given QA pair, the refinement loop terminates when the generated question $q_k$ becomes unsolvable

by the **Less Knowledgeable Peer** ($\mathcal{A}_{\text{LKP}}$), a baseline model formally defined in Stage III, or when a predefined maximum of $K_{\max} = 30$ iterations is reached. This targeted termination ensures that the engine's computational resources are focused on producing problems that precisely challenge the base model's capabilities. In our experiments, the $\mathcal{A}_{\text{LKP}}$ is instantiated as DeepSeek-R1-0528 without tool access.

### C.3 COMPUTATIONAL COST ANALYSIS

We provide a detailed analysis of the computational cost required to generate a single high-quality data point for the $\mathcal{D}_{\text{ZPD}}$ dataset. The cost is broken down into the two primary stages of our pipeline: agentic refinement and MKO verification. All token counts are based on the respective model's tokenizer, and costs are estimated using official API pricing as of the experiment date[1].

#### C.3.1 COST OF AGENTIC REFINEMENT (STAGE II)

In this stage, the refinement agent, $\mathcal{A}_{\text{refine}}$ (DeepSeek-R1), iteratively enhances a QA pair until it reaches the capability frontier of the Less Knowledge Peer (LKP). The cost per data point is variable, depending on the number of iterations ($K$) needed.

On average, processing a single candidate data point involves the following:

- **Refinement Iterations ($K$):** A data point undergoes an average of **7.81** iterations.
- **Token Throughput per API Call:**
  - Input: **18,613.82** tokens.
  - Output: **11,643.22** tokens.
- **Tool Calls per Data Point:**
  - Search: **0.70** calls.
  - Scholar: **0.61** calls.
  - Browser: **1.21** calls (avg. 10,000 tokens/call).
  - Code Interpreter: **0.94** calls (executed locally, no API cost).

**Cost Breakdown.** The average refinement cost per candidate is approximately **\$0.24**, calculated as follows:

- **LLM Cost:** $7.81 \times (18{,}614 \times \$0.56/\text{M} + 11{,}643 \times \$1.68/\text{M}) \approx \$0.234$.
- **Search Cost:** $(0.70 + 0.61) \times \$0.00275/\text{call} \approx \$0.0036$.
- **Browser Cost:** $1.21 \times 10{,}000 \times \$0.00005/\text{token} \approx \$0.0006$.

#### C.3.2 COST OF MKO VERIFICATION (STAGE III)

Candidates that pass the refinement stage are then verified by the More Knowledgeable Other agent, $\mathcal{A}_{\text{MKO}}$ (DeepSeek-V3.1 with tools). This Best-of-N ($N = 3$) verification confirms that the problem is solvable by an expert-level agent, thus ensuring its placement within the Zone of Proximal Development (ZPD).

For the $N = 3$ verification attempts on a single candidate, the average resource consumption is:

- **Total API Calls: 3.32** calls.
- **Token Throughput per API Call:**
  - Input: **20,181.57** tokens.
  - Output: **24,169.88** tokens.
- **Total Tool Calls:**

---

[1]Pricing references: DeepSeek Model API (https://api-docs.deepseek.com/), SerpApi for Google Search (https://serpapi.com/enterprise), and Jina Reader API (https://jina.ai/reader/)

 – Search: **0.50** calls.
 – Scholar: **0.92** calls.
 – Browser: **1.30** calls (avg. 10,000 tokens/call).
 – Code Interpreter: **0.53** calls (executed locally, no API cost).

**Cost Breakdown.**   The verification cost for a single candidate is approximately **\$0.18**:

- **LLM Cost:** $3.32 \times (20,182 \times \$0.56/\text{M} + 24,170 \times \$1.68/\text{M}) \approx \$0.172$.
- **Search Cost:** $(0.50 + 0.92) \times \$0.00275/\text{call} \approx \$0.0039$.
- **Browser Cost:** $1.30 \times 10,000 \times \$0.00005/\text{token} \approx \$0.00065$.

However, only a fraction of candidates pass this stage. With an observed success rate of **33%**, the amortized cost to obtain one successfully verified data point is $\$0.18/0.33 \approx$ **\$0.54**.

In summary, the total end-to-end amortized cost to generate one high-quality, verified PhD-level QA pair with its solution trajectory for $\mathcal{D}_{\text{ZPD}}$ is approximately **\$0.78** (\$0.24 for refinement + \$0.54 for amortized verification). While this represents a non-trivial investment per sample, it aligns with our "quality-over-quantity" approach. This automated pipeline produces a valuable training asset at a fraction of the cost and time that manual curation by human experts would demand.

## C.4   Human Evaluation of Dataset Quality

### C.4.1   Human Review for $\mathcal{D}_{\text{HUMAN}}$

We emphasize that $\mathcal{D}_{\text{human}}$ set is not used for any training. It serves as a diagnostic dataset, composed of samples that our most capable agent (the MKO) failed to solve. The purpose of $\mathcal{D}_{\text{human}}$ is to facilitate in-depth failure analysis, enabling us to understand the limitations of our synthesis engine and to probe the capability frontiers of state-of-the-art (SOTA) agents.

We conducte a qualitative audit on $\mathcal{D}_{\text{human}}$ set by randomly selecting 200 samples from $\mathcal{D}_{\text{human}}$ for manual inspection. For each discipline (e.g., CS, Math, and Biology), the review was performed by three graduate students with relevant expertise. Their task was to diagnose the root cause of the MKO's failure and classify each case into one of three predefined categories:

(A) **Problem Defect**: The problem statement or its ground-truth answer is flawed (e.g., ambiguous, ill-posed, or factually incorrect).

(B) **Execution Gap**: The agent devised a correct high-level plan but failed in its execution (e.g., misinterpreting a retrieved source, overlooking contradictory evidence, or failing to self-correct).

(C) **Strategic Planning Failure**: The agent failed to formulate a viable high-level plan to solve the problem.

Our analysis, summarized in Table 7, reveals that the vast majority of failures stem from agent's intrinsic limitations rather than from data quality issues.

Table 7: Distribution of Failure Modes in $\mathcal{D}_{\text{human}}$

| Failure Type | Percentage (%) |
| --- | --- |
| (A) Problem Defect | 10.5 |
| (B) Execution Gap | 71.0 |
| (C) Strategic Planning Failure | 18.5 |

The predominance of "Execution Gap"(71.0%) is particularly informative. It reveals that even when the MKO can devise a correct strategy, it often fails at the last mile of reasoning. The following case study, involving a specialized structural biology question, illustrates this phenomenon:

**Case Study: An Execution Gap in Structural Biology**

**Question**: What minimum interatomic spacing must exist between the C$\gamma$2 methyl groups of $\beta$-branched residues at position d in coiled-coil hydrophobic cores to prevent steric clashes?

**Agent's Reasoning Trajectory**:

1. **Correct Initial Strategy**: The agent correctly initiated its research using `google scholar` and identified a highly relevant paper (Ramos and Lazaridis, 2011). `google scholar(query="...")`

2. **Successful Information Extraction**: It subsequently employed the `Visit` tool on the paper's PDF, correctly extracting the key quantitative detail: a distance range of 3.6–3.8 Å for valine. `Visit(url="...")`

3. **Failure in Final Synthesis**: Despite possessing the correct information, the agent prematurely concluded its reasoning. It presented 3.6 Å as the final answer, failing to perform a crucial validation step. Specifically, it did not reconcile this value with the ground truth (>5.5 Å), a discrepancy that could have been resolved by considering a different biological context discussed elsewhere in the same paper.

**Analysis:** This case is a clear example of an Execution Gap. The failure was not strategic but tactical, stemming from a lack of critical self-assessment and an over-reliance on the first piece of retrieved information. This underscores the need to advance agent capabilities beyond mere information retrieval towards robust and critical synthesis of retrieved knowledge.

This human-in-the-loop analysis establishes an invaluable feedback loop for our research:

1. **Data Quality Refinement**: It enables us to identify and filter the small fraction of flawed problems (Category A), thereby continuously enhancing the quality of our data synthesis engine in future iterations.

2. **Agent Capability Diagnosis**: More importantly, it provides a detailed qualitative map of the MKO's reasoning deficiencies (Category B) and confirms that our synthesis engine generates data that genuinely challenge the capabilities of SOTA agents (Category C).

### C.4.2 HUMAN REVIEW FOR $\mathcal{D}_{ZPD}$

Ensuring the quality of a fully synthetic dataset is paramount. In addition to flagging unsolvable cases for review, we conducted a rigorous quality control on the final training set. We randomly sampled 200 verified QA pairs from $\mathcal{D}_{ZPD}$ for manual inspection by graduate students with domain expertise. The audit protocol required them to assess each sample against two strict criteria:

- **Problem Quality**: The question must be well-posed, non-trivial, and demonstrably at a postgraduate level of difficulty.

- **Solution Quality**: The agent-generated solution trajectory must be factually correct, logically sound, and exhibit a coherent and valid reasoning process.

The results were highly positive: over 96.5% of the audited samples passed this inspection, confirming the efficacy of our ZPD-based data generation and filtering pipeline.

### C.5 A QUANTITATIVE ANALYSIS OF TASK DIFFICULTY

To provide direct evidence of the diverse cognitive challenges embedded in `AgentFrontier`, we performed a manual analysis to classify the nature of difficulty in our generated tasks. We annotated a random sample of 200 questions from the dataset, categorizing each according to its primary source of difficulty. This process yielded a robust distribution of cognitive demands, as detailed in Table 8.

The results in Table 8 clearly demonstrate that no single difficulty type dominates the dataset. The prevalence of **Quantitative Reasoning** (39.0%) and **Knowledge Fusion** (27.0%) substantiates our claim that `AgentFrontier` moves far beyond tasks solvable by simple information retrieval or linear tool chaining. Instead, it generates complex, realistic research challenges that compel agents to perform multi-faceted reasoning, such as executing code and integrating knowledge across diverse domains.

Table 8: Distribution of primary difficulty types in a random sample of 200 `AgentFrontier` questions. The analysis reveals a balanced composition, with a significant emphasis on reasoning-intensive categories over simple retrieval.

| Difficulty Type | Description | Percentage (%) |
|---|---|---|
| Knowledge Fusion | Requires synthesizing information from multiple, often interdisciplinary, sources to form a coherent conclusion. | 27.0 |
| Quantitative Reasoning | Demands mathematical calculation, logical deduction, or programmatic execution to arrive at a solution. | 39.0 |
| Conceptual Leap | Involves abstracting general principles or theories from concrete examples or identifying non-obvious relationships. | 16.5 |
| Critical Thinking | Necessitates identifying contradictions, evaluating the quality of evidence, or reasoning about anomalies and edge cases. | 17.5 |

In summary, this quantitative analysis, combined with our principled multi-faceted design (Section 2) and the empirical evidence from diverse tool usage (Section 5.2), converges to demonstrate that `AgentFrontier` successfully fosters a rich and realistic spectrum of difficulty.

# D EXPERIMENTAL DETAILS

## D.1 EVALUATION BENCHMARKS

- **HLE** (Phan et al., 2025) - Humanity's Last Exam is an expert-curated benchmark comprising 2,500 challenging questions across a wide range of disciplines, designed to assess frontier-level academic competence. Our evaluation is conducted on its 2,154 text-only questions.

- **ZPD Exam** - Our newly proposed multi-disciplinary benchmark designed to probe the zone of proximal development (ZPD) in LLMs. We use the 1,024 questions from its first version (v1.0).

- **R-Bench** (Guo et al., 2025b) - A graduate-level, multi-disciplinary benchmark designed to assess the complex reasoning capabilities of LLMs. We use its English text-only version. After excluding one question due to potential ambiguity, our evaluation set consists of 1,093 multiple-choice questions.

- **xBench-ScienceQA** (Xbench-Team, 2025) - A curated set of 100 Chinese question-answering items from the xBench suite, designed to evaluate foundational scientific knowledge.

## D.2 BASELINE FINE-TUNING DATASETS

- **TaskCraft** (Shi et al., 2025) - The TaskCraft dataset facilitates the fine-tuning of agent models by programmatically generating agentic tasks at scale. These tasks are characterized by their inclusion of multiple tools, tiered difficulty levels, and verifiable execution trajectories.

- **MegaScience** (Fan et al., 2025) - The MegaScience dataset is constructed by integrating high-quality subsets from multiple open-source scientific datasets to ensure sample abundance and high fidelity. The majority of its questions are sourced from university textbooks.

- **MiroVerse** (MiroMind-Data-Team, 2025) - MiroVerse is an open-source, large-scale dataset for AI agents, covering diverse tasks such as multi-hop question answering, web navigation, and scientific reasoning. We use the SFT data from its v0.1 release.

### D.3 TOOL IMPLEMENTATION

Our agent is equipped with a suite of tools to support its research process, from broad exploration to empirical validation. Each tool is designed for batch processing to enhance efficiency and produces structured outputs for seamless integration into the agent's iterative reasoning loop.

- **Search:** Performs parallel web searches using the Google Search API. It returns a list of structured results, each containing a title, snippet, and URL, allowing the agent to efficiently assess the relevance of multiple sources.

- **Scholar:** Tackles multi-disciplinary challenges by querying the Google Scholar API to navigate scientific literature. It returns structured metadata, including authors, publication venue, and citation counts, enabling the agent to identify authoritative works and their scholarly context.

- **Browser:** Extracts targeted information from a given URL. The agent provides a specific goal (e.g., "extract the dataset and evaluation metrics"). The tool first fetches the page content using Jina Reader (Jina.ai, 2025) and then employs Qwen3 (Yang et al., 2025) to synthesize a precise answer based on the goal. This allows for focused knowledge extraction from web pages.

- **Code:** Provides a sandboxed Python environment for computational analysis and verification. It is equipped with standard scientific libraries (e.g., NumPy, SciPy) and allows the agent to execute code for tasks like data analysis or simulations. All outputs (stdout, stderr, and figures) are captured as text, providing empirical evidence for the agent's reasoning process.

### D.4 TRAINING DETAILS

**CPT Objective.** The continued pre-training (CPT) stage minimizes the standard language modeling loss:

$$\mathcal{L}_{\text{CPT}}(\theta) = -\sum_{t=1}^{T} \log p_\theta(x_t \mid x_{<t}), \tag{1}$$

where $x_t$ denotes the token at position $t$, and $\theta$ are the model parameters.

**RFT Objective.** The rejection sampling fine-tuning (RFT) stage trains the model on accepted research trajectories. Formally, given a research question $q^{(i)}$, the model generates the reasoning report $r_j^{(i)}$ at round $j$ conditioned on the previous report–observation pair $\{r_{j-1}^{(i)}, o_{j-1}^{(i)}\}$, with initialization $r_0^{(i)} = o_0^{(i)} = \emptyset$. For a collection of $K$ accepted trajectories, where trajectory $i$ has $L_i$ rounds, the objective reduces to supervised learning that maximizes the conditional log-likelihood:

$$\mathcal{L}_{\text{RFT}}(\theta) = -\sum_{i=1}^{K} \sum_{j=1}^{L_i} \log p_\theta\left(r_j^{(i)} \,\Big|\, q^{(i)}, r_{j-1}^{(i)}, o_{j-1}^{(i)}\right), \tag{2}$$

where $\theta$ denotes the model parameters. The loss computed is exclusively on the reasoning report tokens; tool observations are included in the context but excluded from backpropagation.

**Implementation.** We implement supervised fine-tuning (SFT) using the Megatron-LM framework (Shoeybi et al., 2019). The hyperparameters for fine-tuning our MoE and Dense models are detailed in Table 9 and Table 10, respectively.

### D.5 ABLATION ON FINE-TUNING DATASETS

Figure 8 presents an ablation study on the impact of different fine-tuning datasets (TaskCraft, Mega-Science, MiroVerse, and our proposed AgentFrontier data) on the performance of Qwen3-8B and Qwen3-32B models. The results, plotted across all four evaluation benchmarks, show that models fine-tuned with our RFT data almost achieve superior performance, highlighting the effectiveness of our data synthesis strategy.

Table 9: SFT Hyperparameters for the MoE Model.

| Parameter | Value |
|---|---|
| Training Epochs | 3 |
| Max Sequence Length | 40,960 |
| Batch Size | 256 |
| Learning Rate | $7.0 \times 10^{-6}$ |
| Learning Rate (Min) | $7.0 \times 10^{-7}$ |
| LR Scheduler | Linear Decay |
| Tensor Parallel (MP) | 4 |
| Expert Parallel (EP) | 2 |
| Pipeline Parallel (PP) | 1 |

Table 10: SFT Hyperparameters for the Dense Model.

| Parameter | Value |
|---|---|
| Training Epochs | 3 |
| Max Sequence Length | 40,960 |
| Batch Size | 64 |
| Learning Rate | $4.0 \times 10^{-5}$ |
| LR Scheduler | Cosine Decay |
| Warmup Ratio | 0.1 |

Table 11 presents a detailed analysis of tool usage and conditional accuracy for Qwen3-30B-A3B model after undergoing rejection-sampling fine-tuning (RFT) on four distinct datasets. The results clearly demonstrate the effectiveness of our synthesized dataset, AgentFrontier. The agent fine-tuned on AgentFrontier achieves the highest overall conditional accuracy on both the ZPD-Exam (87.6%) and RBench-T (63.7%) benchmarks. Furthermore, it consistently secures top-tier accuracy for critical tools across various benchmarks, such as for the Scholar (91.7%) and Browser (91.8%) tools on ZPD-Exam and the Code tool on both ZPD-Exam (83.3%) and RBench-T (78.6%). This superior performance underscores the quality of AgentFrontier in enhancing an agent's capability to correctly and robustly utilize tools across a diverse range of complex tasks.

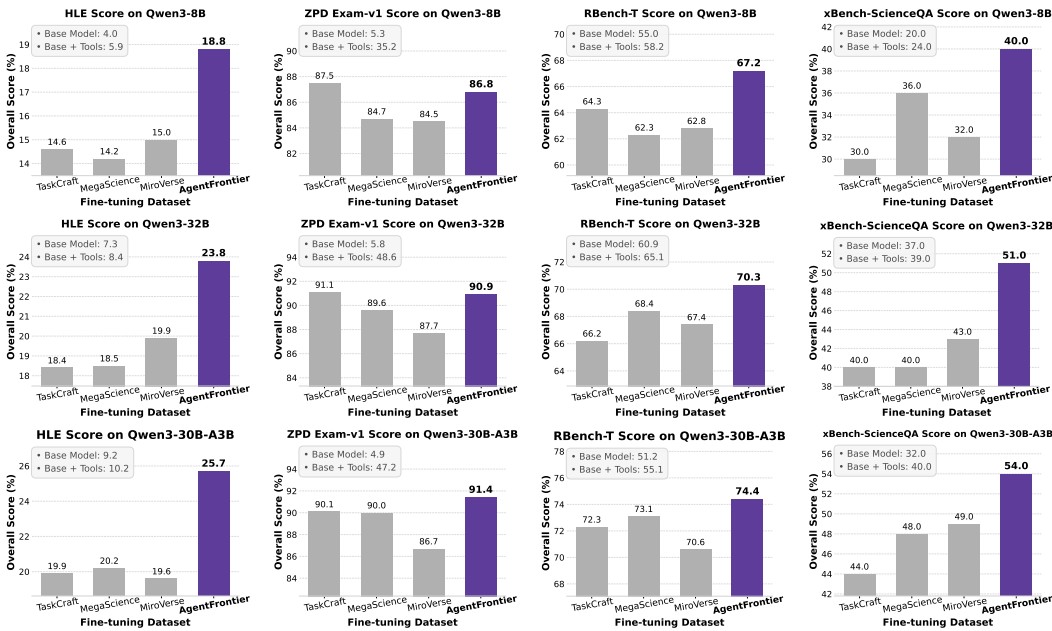

Figure 8: Impact of different fine-tuning datasets on the performance of Qwen3-8B (top row), Qwen3-32B (mid row), and Qwen3-30B-A3B (bottom row) across four evaluation benchmarks.

## D.6 CPT RESULTS AND COMPARISONS

Table 12 provides a comprehensive comparison of our model, AgentFrontier-30B-A3B, with state-of-the-art proprietary and open-source models on the four evaluation benchmarks. We report results

Table 11: Tool usage statistics for the Qwen3-30B-A3B agent on the ZPD Exam, RBench-T and xBench-ScienceQA. Each column block shows performance after RFT on a different dataset. We report average usage per round and conditional tool accuracy (Acc, %), defined as the success rate for tasks that use the tool. The final row details overall metrics. Best results are in **bold**.

| Benchmark | Fine-tuning Dataset
Tool / Metric | TaskCraft | | MegaScience | | MiroVerse | | AgentFrontier | |
|---|---|---|---|---|---|---|---|---|---|
| | | Usage | Acc (%) | Usage | Acc (%) | Usage | Acc (%) | Usage | Acc (%) |
| HLE | Search | 0.68 | 19.6 | 0.67 | 20.3 | **0.73** | 20.4 | **0.73** | **24.9** |
| | Scholar | 0.78 | 21.0 | **0.98** | 20.3 | 0.87 | 20.6 | 0.89 | **25.4** |
| | Browser | 1.24 | 25.2 | 1.39 | 23.4 | **1.47** | 22.7 | 1.32 | **29.8** |
| | Code | 0.52 | 18.1 | 0.65 | 18.6 | **0.67** | 18.4 | 0.63 | **24.9** |
| | **Overall** (Rounds/Acc.) | 4.21 | 21.0 | 4.70 | 20.6 | **4.74** | 20.5 | 4.57 | **26.3** |
| ZPD-Exam | Search | 0.15 | **90.8** | 0.10 | 85.4 | **0.18** | 74.8 | 0.13 | 83.6 |
| | Scholar | 1.20 | 90.1 | **1.28** | 90.2 | 1.22 | 87.3 | 1.23 | **91.7** |
| | Browser | 1.39 | 90.6 | 1.35 | 91.0 | **1.46** | 86.9 | 1.45 | **91.8** |
| | Code | 0.03 | 78.1 | 0.03 | 68.6 | 0.02 | 66.7 | **0.04** | **83.3** |
| | **Overall** (Rounds/Acc.) | 3.77 | 87.4 | 3.76 | 83.8 | **3.88** | 78.9 | 3.84 | **87.6** |
| RBench-T | Search | 0.23 | 55.0 | 0.24 | 53.6 | 0.26 | 50.0 | **0.28** | **58.1** |
| | Scholar | 0.14 | **63.1** | 0.15 | 59.6 | **0.16** | 54.8 | **0.16** | 59.7 |
| | Browser | 0.20 | 54.4 | 0.22 | 53.8 | **0.28** | 46.9 | 0.27 | **58.2** |
| | Code | 0.74 | 77.5 | 0.80 | **78.6** | 0.83 | 77.2 | **0.88** | **78.6** |
| | **Overall** (Rounds/Acc.) | 2.31 | 62.5 | 2.41 | 61.4 | 2.53 | 57.2 | **2.59** | **63.7** |
| xBench-SciQA | Search | **0.44** | 28.6 | 0.39 | 50.0 | 0.36 | 46.4 | 0.43 | **57.1** |
| | Scholar | 0.29 | 54.2 | **0.39** | 44.8 | 0.36 | **66.7** | 0.28 | 48.1 |
| | Browser | 0.46 | 31.6 | **0.61** | 38.5 | 0.48 | **52.4** | 0.36 | 42.1 |
| | Code | **0.62** | 47.2 | 0.54 | 46.8 | 0.60 | 42.6 | 0.58 | **55.6** |
| | **Overall** (Rounds/Acc.) | 2.81 | 40.4 | **2.93** | 45.0 | 2.81 | **52.0** | 2.66 | 50.7 |

for models with and without tool access. The final rows highlight the performance of our model and quantify the significant gains achieved through the Continued Pre-training (CPT) stage.

Table 12: Comparison of AgentFrontier with state-of-the-art proprietary and open-source LLMs/Agents on four high-level multi-disciplinary benchmarks. [†] marks the result from the corresponding official reports. The final row highlights the performance gain from our Continued Pre-training (CPT) stage.

| LLMs/Agents | Tools | HLE (text-only) | ZPD Exam-v1 | RBench-T | xBench-ScienceQA |
|---|---|---|---|---|---|
| *Direct Inference (with and without Tools)* | | | | | |
| GPT-4o | ✗ | 2.3 | 4.8 | 42.0 | 13.0 |
| | ✓ | 4.8 | 51.3 | 48.5 | 15.0 |
| Claude 4 Sonnet | ✗ | 5.4 | 6.0 | 61.8 | 32.0 |
| | ✓ | 14.3 | 86.6 | 71.1 | 47.0 |
| Gemini 2.5 Flash | ✗ | 10.4 | 6.3 | 65.2 | 35.0 |
| | ✓ | 12.6 | 58.1 | 75.8 | 39.0 |
| DeepSeek V3.1-671B | ✗ | 18.5 | 8.2 | 76.3 | 40.0 |
| | ✓ | 29.8[†] | 93.1 | **79.4** | 55.0 |
| Qwen3-30B-A3B (Thinking-2507) | ✗ | 9.2 | 4.9 | 51.2 | 32.0 |
| | ✓ | 10.2 | 47.2 | 55.1 | 40.0 |
| *Proprietary Research Agents* | | | | | |
| OpenAI DeepResearch | ✓ | 26.6[†] | – | – | – |
| Gemini DeepResearch | ✓ | 26.9[†] | – | – | – |
| Kimi-Researcher | ✓ | 26.9[†] | – | – | – |
| *Open-source Agents* | | | | | |
| WebDancer-QwQ-32B | ✓ | 6.4 | 51.8 | 67.6 | 38.0 |
| WebSailor-72B | ✓ | 9.2 | 62.1 | 44.9 | 27.0 |
| WebShaper-72B | ✓ | 8.0 | 54.4 | 66.8 | 29.0 |
| *Ours* | | | | | |
| **AgentFrontier-30B-A3B (RFT only)** | ✓ | 25.7 | 91.4 | 74.4 | 54.0 |
| **AgentFrontier-30B-A3B (CPT+RFT)** | ✓ | 28.6 | **93.4** | 77.1 | **61.0** |
| Δ (CPT gain) | | **+2.9** | **+2.0** | **+2.7** | **+7.0** |

## D.7 ABLATION STUDY ON THE ZPD CALIBRATION POLICY

To isolate the impact of our ZPD-calibration policy, we conduct an ablation study. This experiment directly compares our ZPD-based data selection against a random sampling baseline, to verify that the performance gains are attributable to our targeted calibration strategy rather than a simpler sampling heuristic.

**Experimental Setup.** We fine-tune three models of varying scales (Qwen3-8B, Qwen3-32B, and Qwen3-30B-A3B) on 12,000 trajectories sampled from $D_{\text{refined}}$. The experiment compares two conditions, differing only in data selection method: (1) **ZPD Selection (Ours):** Selecting trajectories via our proposed ZPD-calibration policy. (2) **Random Selection (Baseline):** Randomly sampling an equal number of trajectories from the same pool, $D_{\text{refined}}$.

**Results and Analysis.** The results, presented in Table 13, show that our ZPD-calibration policy. Across all model scales and evaluation benchmarks, fine-tuning on data selected via our ZPD policy consistently and significantly outperforms the random baseline. The most substantial gains are observed on HLE and xBench-SciQA, with improvements of up to +10.0 points. These benchmarks are specifically designed to evaluate deep, multi-step reasoning. This result strongly suggests that our ZPD-based mechanism is not merely a difficulty filter, but a targeted strategy that prioritizes trajectories fostering complex reasoning and knowledge fusion—the core capabilities our work aims to enhance.

Table 13: Ablation study comparing our ZPD-based data selection against a random sampling baseline. Models are fine-tuned on 12,000 trajectories from $D_{\text{refined}}$. Scores are reported on four benchmarks, with the performance delta over the baseline shown in parentheses. Best results are in **bold**.

| Base Model | Data Selection | HLE | ZPD Exam-v1 | RBench-T | xBench-SciQA |
|---|---|---|---|---|---|
| Qwen3-8B | Random Selection | 16.9 | 85.1 | 66.1 | 33.0 |
| | ZPD Selection (Ours) | **18.8** (+1.9) | **86.8** (+1.7) | **67.2** (+1.1) | **40.0** (+7.0) |
| Qwen3-32B | Random Selection | 20.9 | 88.0 | 69.5 | 45.0 |
| | ZPD Selection (Ours) | **23.8** (+2.9) | **90.9** (+2.9) | **70.3** (+0.8) | **51.0** (+6.0) |
| Qwen3-30B-A3B | Random Selection | 20.9 | 89.1 | 72.2 | 44.0 |
| | ZPD Selection (Ours) | **25.7** (+4.8) | **91.4** (+2.3) | **74.4** (+2.2) | **54.0** (+10.0) |

## D.8 HYPERPARAMETER SENSITIVITY ANALYSIS

This section analyzes the sensitivity of two key hyperparameters in our data synthesis pipeline: the Best-of-N (BoN) verification attempts, $N$, and the redundancy threshold, $\epsilon$. Our analysis validates the chosen values by examining the trade-offs among data quality, yield, and computational cost.

### D.8.1 BEST-OF-N VERIFICATION ATTEMPTS ($N$)

The hyperparameter $N$ for Best-of-N (BoN) verification controls the] trade-off between **data yield** and **computational cost**. While a higher $N$ increases the chance of verifying a difficult problem (thus increasing yield), the cost scales linearly with $N$. We quantify this by testing $N \in [1, 8]$ on 1,000 candidate QA pairs. The results are presented in Table 14.

Table 14: Sensitivity analysis for the Best-of-N hyperparameter $N$. We observe diminishing returns in pass@N as $N$ increases, while the cost scales linearly. $N = 3$ is identified as the optimal elbow point, maximizing the gain in yield for the incurred computational cost.

| N | 1 | 2 | 3 | 4 | 5 | 6 | 7 | 8 |
|---|---|---|---|---|---|---|---|---|
| pass@N (%) | 29.5 | 36.0 | 40.0 | 43.2 | 45.1 | 46.8 | 48.2 | 48.9 |
| Marginal Gain (%) | – | +6.5 | +4.0 | +3.2 | +1.9 | +1.7 | +1.4 | +0.7 |
| Relative Cost | 1× | 2× | 3× | 4× | 5× | 6× | 7× | 8× |

As shown in Table 14, the results demonstrate a clear pattern of **diminishing returns**. The yield (pass@N) grows substantially up to $N = 3$ (a 10.5% absolute gain), but the marginal gain shrinks sharply thereafter. For instance, increasing $N$ from 4 to 8 only yields an additional 5.7% of data at double the cost. We therefore identify $N = 3$ **as the optimal elbow point**. This value strikes an effective balance between high data yield and acceptable computational cost.

### D.8.2 REDUNDANCY THRESHOLD ($\epsilon$)

The redundancy threshold, $\epsilon$, balances **dataset diversity** against **data volume**. We chose $\epsilon = 0.7$ based on both quantitative analysis and qualitative inspection.

Table 15: Cumulative percentage of data retained as a function of the similarity threshold $\epsilon$. Setting $\epsilon = 0.7$ filters approximately 30% of the most similar pairs while retaining 70% of the data, striking a balance between diversity and volume.

| Threshold ($\epsilon$) | 0.1 | 0.2 | 0.3 | 0.4 | 0.5 | 0.6 | 0.7 | 0.8 | 0.9 |
|---|---|---|---|---|---|---|---|---|---|
| Retained Data (%) | 5.12 | 13.94 | 24.55 | 36.29 | 48.50 | 59.20 | 70.42 | 81.69 | 93.43 |

**Quantitative Analysis** Setting $\epsilon = 0.7$ retains approximately 70% of the data while filtering the ~30% of pairs most likely to be redundant. As Table 15 shows, a lower threshold (e.g., $\epsilon < 0.7$) would significantly reduce data volume, whereas a higher one (e.g., $\epsilon > 0.7$) would be less effective at enhancing diversity.

**Qualitative Analysis** To validate this threshold, we qualitatively inspected pairs with similarity scores around 0.7. We find it serves as an effective semantic cutoff, distinguishing semantically redundant paraphrases from complementary reasoning problems.

**Case 1: Redundant (Score = 0.796 > $\epsilon$)** These two questions are essentially paraphrases that target the same core concept, offering little additional training value.

```
QA-pair 1:
Question: What property of the partially ordered set of
equivalence classes of subsets of the rationals under
homeomorphic embeddability guarantees the absence of both
infinite antichains and infinite strictly decreasing chains?

Answer: partially well-ordered

QA-pair 2:
Question: For equivalence classes of subsets of Q under
topological embeddability, what binary relation defines the
partial order between distinct equivalence classes [A] and
[B] in the pose+t structure?
Answer: homeomorphic embeddability

Similarity: 0.796
```

**Case 2: Complementary (Score = 0.707 ≈ $\epsilon$)** In contrast, these questions probe different facets (a character vs. the underlying principles) of the same scenario, offering complementary training value.

```
QA-pair 1:
Question: In a 1993 meta-cinematic work, which character's
decision to abort a bicycle stunt--after calculating
97.3% fatality probability through narrative role
analysis--demonstrates correct application of the formula
FΔt = mΔv to avoid momentum conservation violations?
Answer: Danny Madigan
```

```
QA-pair 2:
Question:  In Last Action Hero's bicycle stunt scene, Danny
Madigan's abort decision combines which three elements:  (1)
a narrative trope realization about his character role, (2)
implicit application of a momentum conservation principle,
and (3) avoidance of a high-probability negative outcome
derived from collision physics?
Answer:  comedy sidekick, impulse-momentum theorem, fatal
trauma

Similarity:  0.707
```

In summary, our choice of $\epsilon = 0.7$ is empirically justified, balancing dataset diversity and volume to ensure the final training set is rich in novel reasoning challenges.

## E    CASE STUDY

**Clinical Scenario from Humanity's Last Exam**

**Question:** A 68 year old patient presents to your clinic with ankle pain and swelling. The swelling began after a long walk with his wife the previous day. Physical exam shows erythema, slight edema, and pain on flexion and extension as well as mild bony tenderness. X-rays are negative for any acute abnormality. The patient is treated with indomethacin. He returns to the clinic 10 days later with the same pain, swelling, and redness. Laboratory workup reveals a slightly elevated uric acid and C-reactive protein. X-rays once again rule out any acute findings. The patient is treated with a prednisone taper, but his symptoms begin to worsen. Joint aspiration and synovial fluid analysis reveals no crystals and gram stain reveals no organisms or white blood cells. What is this patient's diagnosis?

---

**Ground Truth Diagnosis: Chronic osteomyelitis**

**✖ OpenAI DeepResearch Output (Incorrect)**

Based on the clinical presentation and diagnostic findings, this patient's most likely diagnosis is **Charcot Arthropathy**. Here's the reasoning:

1. Chronic and Recurrent Swelling with Erythema: Charcot arthropathy often presents with recurrent swelling, warmth, and erythema, which can be mistaken for infections or inflammatory arthritis.

2. X-rays Negative for Acute Findings: Early Charcot arthropathy may not show significant abnormalities on X-rays, though later stages may demonstrate joint destruction or deformity.

3. Joint Aspiration Findings: The absence of crystals (ruling out gout or pseudogout) and no organisms or WBCs (ruling out septic arthritis) suggests a non-infectious etiology.

4. Risk Factors: This is commonly associated with diabetes mellitus or other neuropathic conditions, which should be explored in this patient.

*Research completed in 10m · 20 sources · 75 searches*

**Why Charcot Arthropathy Fits Best**

Several clinical clues in this case strongly support Charcot neuropathic arthropathy as the correct diagnosis:

- **Normal X-rays despite severe symptoms:** In early Charcot arthropathy (Stage 0), patients have an acutely inflamed joint before any bony destruction is visible on radiographs. This patient's X-ray showed no acute abnormalities even though his ankle was red, hot, and swollen – a classic scenario for early Charcot changes...

- **Peripheral neuropathy signs and risk factors:** Charcot arthropathy virtually always occurs in the setting of neuropathy. Although not explicitly stated, a 68-year-old patient is likely to have neuropathy...

- **Laboratory and fluid studies exclude other causes:** The synovial fluid had **no crystals, no organisms, and minimal inflammatory cells,** which is exactly what one might find in Charcot...

- **Episode triggered by activity (trauma):** The history that symptoms recur "following activity" aligns with Charcot pathogenesis. Episodes often start after minor trauma or increased weight-bearing exercise...

- **Initial improvement with NSAIDs, then worsening on steroids:** In Charcot arthropathy, treating with just anti-inflammatories can temporarily reduce swelling and pain, but if the patient continues to walk on the joint, the underlying bony collapse progresses...

In summary, the constellation of a relatively **painless, swollen red ankle**, with **normal X-rays**, **negative joint fluid**, and a history of neuropathy or diabetes, is virtually pathognomonic for **Charcot neuropathic arthropathy** once infection and crystals are ruled out. The case matches this picture closely, making Charcot joint the most likely diagnosis.

**Sources:**

1. Amelia Harris, Melanie Violand. Charcot Neuropathic Osteoarthropathy. StatPearls, 2023

2. Botek G, Hild G. Differential diagnosis of Charcot arthropathy. Lower Extremity Review. 2011

3. Pinzur MS. Charcot foot: clinical clues, diagnostic strategies, and treatment principles. American Family Physician. 2018;97(9):594-600

4. Rosskopf AB, et al. The Charcot foot: a pictorial review. Insights Imaging. 2019;10(1):77

5. Singh D, et al. Diagnosing acute monoarthritis in adults: a practical approach. Am Fam Physician. 2003;68(1):83-90

6. Lower Extremity Review (2011). Charcot arthropathy imaging and management

7. Radiopaedia. Osteoarthritis of the ankle – radiographic features

8. Lower Extremity Review (2011). Charcot foot vs. gout and infection

---

✅ **Our Model: AgentFrontier-30B-A3B (Correct)**

---

### Round 1: Initial Analysis & Planning

**Status Report and Deep Analysis   Problem Essence Analysis:** The patient is a 68-year-old presenting with acute ankle pain, swelling, erythema, and tenderness following a long walk... The paradoxical worsening with prednisone is particularly notable and may point to a condition where steroids exacerbate symptoms, such as certain infections or rare inflammatory disorders...

**Problem-Solving Planning:** The analysis confirms that common diagnoses are unlikely. However, chronic osteomyelitis or rare conditions like tuberculous arthritis remain possibilities... Further verification through academic sources is needed to identify conditions that match this specific presentation, especially those where prednisone worsens symptoms.

**Tool Call:**

```
{"name": "google_scholar", "arguments": {"query": ["ankle
    pain swelling erythema no crystals no organisms elevated
    uric acid", "differential diagnosis ankle arthritis
    negative synovial fluid", "prednisone worsens arthritis
    no crystals"]}}
```

---

### Round 2: Synthesizing Evidence & Refining Hypothesis

**Status Report and Deep Analysis   Progress Achieved:** Based on the initial clinical presentation and subsequent diagnostic findings, the following comprehensive information has been collected...

- **Differential Diagnosis Analysis**: Osteoarthritis (Unlikely), Charcot Arthropathy (Unlikely due to no mention of neuropathy), Septic Arthritis (Ruled out), Pseudogout (Ruled out), **Chronic Osteomyelitis (Possible)**.
- **Key Diagnostic Challenges**: Paradoxical worsening with prednisone therapy suggests an infection that is exacerbated by immunosuppression...
- **Literature Findings from Tool Response**: Search results indicate that certain infections (particularly tuberculous or fungal arthritis) can present with negative synovial fluid cultures and gram stains... Chronic osteomyelitis can present with intermittent symptoms and minimal radiographic changes.

**Next Steps Plan:** Given the complex presentation... the next step should focus on obtaining more specific diagnostic information about chronic osteomyelitis presentations and its differentiation from other conditions.

**Tool Call:**

```
{"name": "google_scholar", "arguments": {"query": ["chronic
    osteomyelitis ankle differential diagnosis", "
    osteomyelitis ankle negative synovial fluid", "ankle
    arthritis steroid exacerbation", "tuberculous arthritis
    ankle presentation"]}}
```

---

**Round 3: Finalizing Diagnosis**

**Status Report and Deep Analysis  Progress Achieved:** The second, more targeted literature search provided crucial confirmatory evidence, allowing for a final synthesis of all findings and a resolution of the remaining diagnostic uncertainties.

- **Synthesized Literature Findings**: The model integrated results from the second tool call to build a robust case for chronic osteomyelitis.

  - **Explaining Negative Findings**: The search for "osteomyelitis ankle negative synovial fluid" yielded articles confirming that false negative results in synovial fluid cultures and gram stains are possible, and that "culture-negative osteomyelitis" is a recognized clinical entity. This directly addressed a key uncertainty from Round 1, explaining why a bone infection could be present despite "clean" joint fluid. Similarly, literature confirmed that initial radiographs are often negative in osteomyelitis, validating the patient's repeated negative X-rays.
  - **Confirming the Steroid Paradox**: The search on "ankle arthritis steroid exacerbation" reinforced the understanding that corticosteroid use can lead to adverse outcomes. By cross-referencing this with the infectious nature of osteomyelitis, the model concluded that the patient's worsening symptoms were a classic sign of an occult infection being unmasked by immunosuppression, transforming this paradoxical finding into a powerful diagnostic clue.
  - **Differentiating from Alternatives**: The search on "tuberculous arthritis ankle presentation" revealed that it typically presents as a subacute or chronic monoarthritis, often over weeks to months. This contrasts with the patient's more acute onset following a specific event (the long walk), making tuberculous arthritis a less likely, though still possible, alternative compared to chronic osteomyelitis.

- **Re-evaluation of Clinical Data**: In light of the new evidence, the model re-evaluated the initial data with greater clarity.

  - The **bony tenderness** is now seen as a specific sign of bone involvement, not just generalized joint pain.
  - The elevated **C-reactive protein (CRP)** is confirmed as a marker of significant inflammation, consistent with a bone infection.
  - The slightly elevated **uric acid** is confidently dismissed as a non-specific finding or a red herring, as no other evidence supports a diagnosis of gout.

- **Diagnostic Certainty Assessment**: The model's confidence in the diagnosis of Chronic Osteomyelitis increased substantially. All other possible diseases had been systematically ruled out by clinical data. Chronic Osteomyelitis remained the only diagnosis that could cohesively explain the entire constellation of findings: persistent symptoms, bony tenderness, negative initial imaging and fluid analysis, and the critical paradoxical response to steroid therapy. The supporting literature provided a solid foundation for this conclusion.

**Next Steps Plan:** No further investigation is required. The diagnostic pathway has been conclusively established through systematic elimination of alternatives and the integration of clinical findings with targeted, confirmatory literature evidence. The available information is sufficient to provide a definitive diagnosis.

---

### Final Answer and Clinical Reasoning

Based on the comprehensive clinical evaluation and integration of supporting literature evidence, this patient's diagnosis is **Chronic Osteomyelitis**.

DIAGNOSTIC PROCESS

The diagnostic journey for this 68-year-old patient began with acute ankle pain and swelling... The most critical diagnostic turning point came with the synovial fluid analysis... combined with the patient's paradoxical worsening during prednisone therapy...

DIFFERENTIAL DIAGNOSIS ELIMINATION

Through systematic exclusion of alternative diagnoses, chronic osteomyelitis emerges as the most consistent explanation...

PATHOPHYSIOLOGICAL MECHANISMS SUPPORTING CHRONIC OSTEOMYELITIS

Chronic osteomyelitis provides the most coherent explanation for this clinical scenario through several key mechanisms:

**Insidious Presentation with Normal Radiographs**  Chronic osteomyelitis often presents with minimal early radiographic changes...

**Paradoxical Response to Steroids**  The worsening during prednisone therapy is particularly telling. Corticosteroids suppress the immune system, allowing previously contained bacterial colonies within necrotic bone to proliferate unchecked...

**Synovial Fluid Findings**  The negative synovial fluid analysis does not exclude osteomyelitis, as the infection originates in the bone marrow...

**Bony Tenderness Without Acute Trauma**  The mild bony tenderness... indicates underlying bone pathology consistent with osteomyelitis...

SUPPORTING EVIDENCE FROM LITERATURE

Recent medical literature corroborates this diagnostic conclusion. Studies indicate that chronic osteomyelitis of the foot and ankle frequently presents with challenging diagnostic features... The paradoxical worsening with corticosteroid therapy aligns perfectly with the known pathophysiology...

CONCLUSION

In summary, this patient's clinical presentation—characterized by acute inflammatory symptoms, normal radiographs, negative synovial fluid analysis, and paradoxical worsening with steroid therapy—best aligns with chronic osteomyelitis rather than any of the other listed conditions...

# F PROMPTS USED IN EXPERIMENTS

The key prompts used in our experiments are presented below to ensure reproducibility.

## F.1 EVALUATION PROMPT

---

**Evaluation Prompt**

Judge whether the following `[response]` to `[question]` is correct or not based on the precise and unambiguous `[correct_answer]` below.

**[question]**: `{question}`
**[response]**: `{response}`

Your judgement must be in the format and criteria specified below:

**`extracted_final_answer`:** The final exact answer extracted from the `[response]`. Put the extracted answer as `'None'` if there is no exact, final answer to extract from the response.

**`[correct_answer]`:** `{correct_answer}`

**`reasoning`:** Explain why the `extracted_final_answer` is correct or incorrect based on `[correct_answer]`, focusing only on if there are meaningful differences between `[correct_answer]` and the `extracted_final_answer`. Do not comment on any background to the problem, do not attempt to solve the problem, do not argue for any answer different than `[correct_answer]`, focus only on whether the answers match.

**`correct`:** Answer `'yes'` if `extracted_final_answer` matches the `[correct_answer]` given above, or is within a small margin of error for numerical problems. Answer `'no'` otherwise, i.e. if there if there is any inconsistency, ambiguity, non-equivalency, or if the extracted answer is incorrect.

**`confidence`:** The extracted confidence score between `0|%|` and `100|%|` from `[response]`. Put `100` if there is no confidence score available.

---

## F.2 SIMILARITY FILTER PROMPT

---

**Similarity Filter Prompt**

Determine if the candidate QA pair expresses **EXACTLY** the same specific question and answer as the reference QA pair.

**Requirements:**

1. The question must ask for identical information with identical technical requirements.

2. The answer must provide identical content with identical technical details.

3. Any difference in the specific information requested or provided means they are NOT identical.

4. Pay special attention to mathematical expressions, symbols, and technical specifications.

---

## F.3 AGENTIC REFINEMENT PROMPT

---

**Prompt for Agentic Refinement ($\mathcal{A}_{\text{refine}}$)**

**Role and Objective:**
You are a sophisticated agent tasked with iterative data refinement. Your primary mission is to transform a given Question-Answer pair $(q_k, a_k)$ into a more complex, in-depth, and factually grounded pair $(q_{k+1}, a_{k+1})$. This escalation must be achieved by leveraging a specialized tool suite $\mathcal{T} = \{T_{\text{search}}, T_{\text{scholar}}, T_{\text{browser}}, T_{\text{code}}\}$.

**Input:**
The current QA pair QA pair $(q_k, a_k)$ in a structured format.

**Mandatory Refinement Protocol:**
Your task is to generate a new, superior QA pair by applying one or more of the following four refinement dimensions. For each generated pair, you **must** utilize the provided tools and explicitly log their usage.

1. **Knowledge Expansion:**
   - **Objective:** Broaden the informational scope of the QA pair.
   - **Action:** You **must** use the $T_{\text{search}}$, $T_{\text{scholar}}$, or $T_{\text{browser}}$ tools to discover and retrieve relevant background knowledge, historical context, or contrasting perspectives.
   - **Implementation:** Weave this new information seamlessly into the refined question $(q_{k+1})$ and provide a comprehensive explanation in the refined answer $(a_{k+1})$.

2. **Conceptual Abstraction:**
   - **Objective:** Elevate the level of abstract reasoning required.
   - **Action:** Analyze the core concepts within $(q_k, a_k)$. Formulate a new question $(q_{k+1})$ that requires identifying higher-level principles, synthesizing information to uncover subtle relationships, or drawing non-obvious analogies.
   - **Implementation:** The refined answer $(a_{k+1})$ must explicitly articulate this abstract principle or relationship. You may use $T_{\text{scholar}}$ to find established theoretical frameworks to aid this process.

3. **Factual Grounding:**
   - **Objective:** Enhance the factual accuracy, precision, and verifiability.
   - **Action:** You **must** use $T_{\text{search}}$ and $T_{\text{scholar}}$ to perform multi-source cross-validation of the facts and claims in $a_k$.
   - **Implementation:** Augment the refined answer $(a_{k+1})$ with precise quantitative data, specific named entities, and direct citations or references to the authoritative sources you retrieved.

4. **Computational Formulation:**
   - **Objective:** Introduce a verifiable computational or logical reasoning challenge.
   - **Action:** You **must** use the $T_{\text{code}}$ tool (a Python execution environment) to design a new question $(q_{k+1})$ that necessitates a quantitative calculation or algorithmic simulation.
   - **Implementation:** The refined answer $(a_{k+1})$ must contain: (1) The complete, executable Python code block used to solve the problem, and (2) The final output produced by the code, along with a brief explanation.

**Tool Usage Protocol:** {tools}

**Final Instruction:**
Proceed with the refinement of the provided $(q_k, a_k)$. Your response must be only the final JSON object.

---

