# OpenReview forum: "Expanding the Capability Frontier of LLM Agents with ZPD-Guided Data Synthesis"
_ICLR.cc/2026/Conference — ICLR 2026 Poster_

### Official Review · Reviewer_eHCL · 2025-10-24

**Soundness:** 4
**Presentation:** 4
**Contribution:** 3
**Rating:** 8
**Confidence:** 3

**Summary:**

This paper introduces a novel framework for data synthesis and evaluation guided by the Zone of Proximal Development (ZPD) principle from educational psychology. The authors propose the AgentFrontier Engine, a multi-agent pipeline that automatically generates, refines, and filters reasoning tasks that lie just beyond a model’s current competence but within its learnable range. By distinguishing between what a base model (LKP) can solve independently and what a stronger agent (MKO) can solve with tools, the framework creates a dynamic “ZPD dataset” that pushes model capabilities systematically. The paper also presents ZPD Exam, an evolving benchmark derived from these tasks. Experiments on Qwen and other models demonstrate significant performance gains across reasoning and tool-augmented benchmarks, suggesting that ZPD-guided synthesis effectively expands model reasoning frontiers.

**Strengths:**

This paper introduces a compelling theoretical grounding (ZPD) for guiding data generation, a fresh and rigorous way to define model capability boundaries. The model trained with the data generated by the designed pipline outputperfom state-of-the-art results on
demanding benchmarks. The authors denote that their data and model will be public, which should be a great source to advance the deep research ability of public models. Overall, I think this is a good paper.

**Weaknesses:**

1. The paper lack analysis on how to config the MKO and LKP.
2. Whlst the paper says "ZPD-guided synthesis continually redefines the capability frontier based on the model’s evolving competence", the pipline to generate the data is complicated and might not easy to implement.

**Questions:**

See Weaknesses

---

> ### Author Response · Authors · 2025-11-20
> **Response to Reviewer eHCL (Part 1/1)**
>
> Dear Reviewer eHCL,
>
> We are very grateful for your thorough evaluation and encouraging feedback. Your constructive review has been invaluable for strengthening our paper. We have carefully considered your comments and have revised our manuscript accordingly, including a new ablation study to address your concerns.
>
> ---
> ### **W1: Analysis on config of the MKO and LKP**
> Thank you for this excellent suggestion. To clarify the principles for configuring the MKO (More Knowledgeable Other) and LKP (Less Knowledgeable Peer), we have conducted a new ablation study. Our findings empirically validate our framework's design principle: the LKP/MKO configuration is governed by a crucial trade-off between **data synthesis efficiency** (yield rate) and **data complexity** (reasoning depth and tool diversity).
>
> **Experimental Setup:** We tested three LKP/MKO configurations on a 1,000-sample subset of our $D_{refined}$ dataset. For context, DeepSeek-V3.1 has inherently stronger reasoning and agentic abilities than its predecessor DeepSeek-R1. [1]
> 1. Original (Balanced Gap)
>     - LKP: DeepSeek-R1 (no tools)
>     - MKO: DeepSeek-V3.1 (with tools)
> 2. Weaker LKP (Wider Gap)
>     - LKP: Qwen3-30B-A3B (no tools)
>     - MKO: DeepSeek-V3.1 (with tools)
> 3. Narrower Gap (Tool-Only Gap)
>     - LKP: DeepSeek-R1 (no tools)
>     - MKO: DeepSeek-R1 (with tools)
>
> **Results:** The results clearly illustrate the trade-off.
>
> Configuration (LKP / MKO)|ZPD Data Yield (%)|Avg. Rounds|Avg. Tool Calls|Tool Usage Distribution (Search/Scholar/Browser/Code)
> -|-|-|-|-
> **1. DS-R1 / DS-V3.1+T (Original)**|**33.1**|**3.32**|**2.32**|0.32 / 0.66 / 0.82 / 0.52
> **2. Qwen3-30B / DS-V3.1+T (Weaker LKP)**|47.7 (↑44%)|1.85 (↓44%)|0.85 (↓63%)|0.18 / 0.23 / 0.36 / 0.08 (all ↓)
> **3. DS-R1 / DS-R1+T (Narrower Gap)**|24.0 (↓27%)|2.99 (≈)|1.99 (≈)|0.19 / 0.67 / 0.58 / 0.55
>
> *ZPD Data Yield Rate = Number of valid $D_{ZPD}$ samples / Total candidate samples.*
>
> **Analysis:**
> 1. **Wider Gap (Config 2)**: Using a weaker LKP (Qwen3-30B) widens the ZPD, significantly increasing data yield (↑44%). However, this efficiency comes at the cost of complexity: the synthesized data involves substantially fewer reasoning rounds (↓44%) and tool calls (↓63%). This produces simpler data that is less effective for training models at the capability frontier.
> 2. **Narrower Gap (Config 3)**: Isolating the gap to tool usage alone results in a sharp drop in yield (↓27%). While the data remains complex (Avg. Rounds/Calls are comparable to original), synthesis process becomes far less efficient.
> 3. **Conclusion**: Our chosen configuration (DS-R1 / DS-V3.1+T) strikes an effective balance. ensuring a scalable data yield while synthesizing data complex enough to advance the capabilities of SOTA models, as validated by our main results.
>
> We have added this full ablation study to Section 5.1. We believe it makes the principles guiding LKP/MKO selection explicit and substantially strengthens the paper.
>
> ---
> ### **W2: Synthetic Data Pipeline**
> We thank you for this crucial observation. The sophistication of our AgentFrontier Engine is indeed a key feature of our work. This complexity is essential for synthesizing data that truly resides at the frontier of LLMs' capabilities. We appreciate this opportunity to elaborate on both the design rationale and our concrete steps to ensure its accessibility to the research community.
>
> **1. Reproducibility and Open-Sourcing.** To directly address the valid concern about implementation difficulty, we have made reproducibility a cornerstone of our contribution. As stated in our Reproducibility Statement, we are committed to a comprehensive open-source release, including:
> - The complete AgentFrontier dataset and the ZPD Exam benchmark.
> - The trained weights of our SOTA model, AgentFrontier-30B-A3B.
> - All prompts, hyperparameters, and detailed setup instructions (Appendices C, D, F).
>
> This release is designed to facilitate replication and extension of our work by the community.
>
> **2. Principled Complexity as a Contribution.** The pipeline's sophistication is a deliberate design choice that operationalizes the ZPD theory into a scalable system. It addresses a crucial trade-off: **scalability vs. cost**. As detailed in our transparent cost analysis (Appendix C.3), our automated pipeline generates expert-level reasoning data at a fraction of the cost (~$0.78/sample) and time required for manual curation by human experts. Therefore, the pipeline's complexity is the necessary investment for a scalable method to advance AI reasoning. We contend that this reproducible blueprint is a key contribution for pushing AI frontiers.
>
> **Reference**
>
> [1] DeepSeek-V3.1: https://api-docs.deepseek.com/news/news250821
>
> ---
> Thank you once again for your encouraging feedback and insightful questions. We hope our responses and the additional experiments have fully addressed your concerns.
>
> Best regards,
>
> Submission5191 Authors

---

> ### Comment · Reviewer_eHCL · 2025-11-25
>
> Thanks for your responses. They make sense to me. Good Luck!

---

> ### Author Response · Authors · 2025-11-26
>
> Dear Reviewer eHCL,
>
> Thank you very much for your positive confirmation and your kind wishes. We are delighted to hear that our explanations were clear and addressed your questions.
>
> Your constructive feedback has been invaluable in strengthening our work, and we truly appreciate your time and support throughout this process.
>
> Best regards,
>
> Submission5191 Authors

---

### Official Review · Reviewer_9U8J · 2025-10-28

**Soundness:** 2
**Presentation:** 2
**Contribution:** 2
**Rating:** 4
**Confidence:** 3

**Summary:**

This work presents a novel data synthesis framework grounded in the Zone of Proximal Development (ZPD). It targets tasks at the frontier of a model’s ability: those that the base model fails at alone but that a stronger, tool-using agent can solve.The framework produces a targeted training dataset and living benchmark to enhance and evaluate agentic reasoning. Training on this dataset boosts performance on various agentic benchmarks and enhances tool use.

**Strengths:**

1) The work offers a clear rule to select data (i.e. when LKP fails in all N=3 unaided attempts while the MKO succeeds). The work validates the approach with experiments.
2) A self-evolving benchmark seems useful, especially given the rate of saturation of existing static benchmarks.

**Weaknesses:**

1) The paper uses just one LKP/MKO pair with LKP=deepseek-r1 and MKO=deepseek-v3.1+tools (apologies if I am mistaken). Thus, it is unclear how general the results are under different LKP and/or MKO.
2) The paper attributes observed gains to ZPD-calibration strategy, but doesn’t directly verify the mechanism. It might be helpful to ablate the calibration policy (ie one could plasibly show how heuristic of random selection compares to the ZPD-calibration). Additionally, it would be nice to see how sensitive the overall framework is to e.g., BoN N and to redundancy threshold $\epsilon$.

**Questions:**

1) How do results change if you swap the LKP for a different model? Or if you use LKO+tools as the MKO instead of a strictly stronger model?
2) How sensitive are the results to BoN N and to the redundancy threshold $\epsilon$?

---

> ### Author Response · Authors · 2025-11-20
> **Response to Reviewer 9U8J (Part 1/2)**
>
> Dear Reviewer 9U8J,
>
> We are grateful for your constructive feedback, which is invaluable in improving our work. Following your suggestions, we conducted targeted new experiments that address your concerns and provide a more comprehensive validation of our ZPD-guided framework.
>
> ---
> ### **W1 & Q1: Different MKO and LKP**
> As you insightfully suggested, the choice of LKP and MKO is critical to the generality of our framework. To rigorously investigate this, we conducted a new ablation study exploring different LKP/MKO capability gaps. The results empirically confirm that our method's effectiveness is governed by a principled trade-off between **data synthesis efficiency** (yield rate) and **data complexity** (reasoning depth).
>
> **Experimental Setup:** We tested 3 LKP/MKO configurations on a 1,000-sample subset of our $D_{refined}$ dataset. For context, DeepSeek-V3.1 has inherently stronger reasoning and agentic abilities than its predecessor DeepSeek-R1.[1] The configurations directly address your questions about using a different LKP and a less powerful MKO:
> 1. Original (Balanced Gap)
>     - LKP: DeepSeek-R1 (no tools)
>     - MKO: DeepSeek-V3.1 (with tools)
> 2. Weaker LKP (Wider Gap)
>     - LKP: Qwen3-30B-A3B (no tools)
>     - MKO: DeepSeek-V3.1 (with tools)
> 3. Narrower Gap (Tool-Only Gap)
>     - LKP: DeepSeek-R1 (no tools)
>     - MKO: DeepSeek-R1 (with tools)
>
> **Results:** Our findings clearly illustrate the trade-off.
>
> Configuration (LKP / MKO)|ZPD Data Yield(%)|Avg. Rounds|Avg. Tool Calls|Tool Usage Distribution (Search/Scholar/Browser/Code)
> -|-|-|-|-
> **1. DS-R1 / DS-V3.1+T (Original)**|**33.1**|**3.32**|**2.32**|0.32 / 0.66 / 0.82 / 0.52
> **2. Qwen3-30B / DS-V3.1+T (Weaker LKP)**|47.7 (↑44%)|1.85 (↓44%)|0.85 (↓63%)|0.18 / 0.23 / 0.36 / 0.08 (all ↓)
> **3. DS-R1 / DS-R1+T (Narrower Gap)**|24.0 (↓27%)|2.99 (≈)|1.99 (≈)|0.19 / 0.67 / 0.58 / 0.55
>
> *ZPD Data Yield Rate = Number of valid $D_{ZPD}$ samples / Total candidate samples.*
>
> **Analysis:**
> 1. **Wider Gap (Config 2)**: Using a weaker LKP (Qwen3-30B) widens the ZPD, significantly increasing data yield (↑44%). However, this efficiency comes at the cost of complexity: the synthesized data involves substantially fewer reasoning rounds (↓44%) and tool calls (↓63%). This produces simpler data that is less effective for training models at the capability frontier.
> 2. **Narrower Gap (Config 3)**: As you suggested, isolating the gap to tool usage alone, results in a sharp drop in yield (↓27%). While the data remains complex (Avg. Rounds/Calls are comparable to original), generation process becomes far less efficient.
> 3. **Conclusion**: These experiments empirically validate that our chosen configuration (DS-R1 / DS-V3.1+T) strikes a critical balance. It maintains a scalable data yield while ensuring the synthesized data is complex enough to push the capabilities of SOTA models, as evidenced by our main results.
>
> We believe this new study makes the principles guiding our LKP/MKO selection explicit and significantly strengthens our paper. We have added this full ablation study to Section 5.1.
>
> ---
> ### **W2: Ablation on Calibration Policy**
> To directly validate that our ZPD-calibration policy is the key driver of performance gains, rather than a simpler sampling heuristic, we performed the exact ablation study you proposed. We compared our ZPD-based selection against a random selection baseline.
>
> **Experimental Setup:** We fine-tuned three different base models (Qwen3 8B, 32B, and 30B-A3B) on 12,000 trajectories from $D_{refined}$. The only variable was the selection method::
> 1. **ZPD Selection (Ours)**: Selected by our ZPD-calibration policy.
> 2. **Random Selection (Baseline)**: Randomly sampled from the same source.
>
> **Results and Analysis**: The results below decisively confirm the superiority of our ZPD-calibration policy.
>
> Base Model|Data Selection Method|HLE|ZPD Exam-v1|RBench-T|xBench-SciQA
> -|-|-|-|-|-
> Qwen3-8B|random selection|16.9|85.1|66.1|33.0
> ||ZPD selection (ours)|**18.8** (+1.9)|**86.8** (+1.7)|**67.2** (+1.1)|**40.0** (+7.0)
> |Qwen3-32B|random selection|20.9|88.0|69.5|45.0
> ||ZPD selection (ours)|**23.8** (+2.9)|**90.9** (+2.9)|**70.3** (+0.8)|**51.0** (+6.0)
> Qwen3-30B-A3B|random selection|20.9|89.1|72.2|44.0
> ||ZPD selection (ours)|**25.7** (+4.8)|**91.4** (+2.3)|**74.4** (+2.2)|**54.0** (+10.0)
>
> Our ZPD-selected data yields **consistently and significantly better performance** across all model scales and benchmarks. Notably, the most substantial gains are on HLE and xBench-SciQA (up to +10.0 points), benchmarks renowned for demanding deep, multi-step reasoning. This provides compelling evidence that our ZPD mechanism is not merely a difficulty filter, but a targeted strategy that effectively identifies trajectories which cultivate the complex reasoning and knowledge fusion capabilities central to our paper's thesis.
>
> Thank you again for this constructive feedback. This ablation materially strengthens our paper's core claims, and we have added it to Appendix D.7.

---

> ### Author Response · Authors · 2025-11-20
> **Response to Reviewer 9U8J (Part 2/2)**
>
> ### **W2 & Q2: Setting of BoN and Redundancy Threshold**
> We appreciate your question regarding hyperparameter sensitivity, which is crucial for establishing the framework's robustness and practicality. We have conducted sensitivity analyses for the two key hyperparameters in our data synthesis pipeline: the Best-of-N verification attempts N and the redundancy threshold $\epsilon$. These analyses justify our choices by demonstrating a principled trade-off between data quality, yield, and computational cost.
> #### 1. BoN N
> The N in Best-of-N (BoN) verification governs the fundamental trade-off between **data yield and computational cost**. A higher N increases the chance of the MKO agent verifying a challenging problem (increasing yield), but at a linearly increasing generation cost. We quantified this trade-off on 1,000 candidate QA pairs, varying N from 1 to 8.
> N|1|2|3|4|5|6|7|8
> -|-|-|-|-|-|-|-|-
> pass@N(%)|29.5|36.0|40.0|43.2|45.1|46.8|48.2|48.9
> Marginal Gain(%)|-|+6.5|+4.0|+3.2|+1.9|+1.7|+1.4|+0.7
> Relative Cost|1x|2x|3x|4x|5x|6x|7x|8x
>
> The results show a clear trend of **diminishing returns**. The yield increases substantially from N=1 to N=3 (a 10.5% absolute gain), but the marginal gain drops sharply thereafter. For instance, doubling the effort from N=4 to N=8 only yields an additional 5.7% of data.
>
> This empirical analysis identifies **N=3 as the clear "elbow point"**. It captures a large portion of verifiable ZPD data without incurring the prohibitive costs of higher N values, ensuring our data synthesis pipeline is both effective and scalable.
> #### 2. Redundancy Threshold $\epsilon$
> Our choice of $\epsilon=0.7$, is a principled decision, not an arbitrary one, grounded in both quantitative analysis of the data distribution and qualitative validation of its semantic impact. It is critical for balancing **dataset diversity** and **data volume**.
>
> **a. Quantitative Analysis**: We analyzed the similarity score distribution across all candidate QA pairs. The cumulative distribution is shown below:
>
> Threshold $\epsilon$ <|0.1|0.2|0.3|0.4|0.5|0.6|0.7|0.8|0.9
> -|-|-|-|-|-|-|-|-|-
> Retained Data Percentage(%)|5.12|13.94|24.55|36.29|48.50|59.20|70.42|81.69|93.43
>
> Setting $\epsilon=0.7$ strikes a deliberate balance: it filters out the top ~30% most similar pairs to enhance diversity while retaining a substantial ~70% of the generated data. A more aggressive threshold (e.g., $\epsilon=0.6$) would discard over 40% of the data, while a more lenient one (e.g., $\epsilon=0.8$) would be less effective at preventing redundancy.
>
> **b. Qualitative Analysis**: To ground our quantitative choice, we inspected pairs on both sides of the threshold. Human validation confirms that $\epsilon=0.7$ acts as a **"semantic cliff"**, distinguishing mere paraphrases from distinct, complementary reasoning challenges.
>
> **Case 1: Redundant (Score=0.796>$\epsilon$)**
>
> These questions are essentially paraphrases targeting the same core concept.
> ```
> QA-pair 1: Question: What property of the partially ordered set of equivalence classes of subsets of the rationals under homeomorphic embeddability guarantees the absence of both infinite antichains and infinite strictly decreasing chains?\nAnswer: partially well-ordered
>
> QA-pair 2: Question: For equivalence classes of subsets of ℚ under topological embeddability, what binary relation defines the partial order between distinct equivalence classes [A] and [B] in the pose+t structure?\nAnswer: homeomorphic embeddability
>
> Similarity: 0.796
> ```
> **Case 2: Complementary (Score=0.707≈$\epsilon$)**
>
> These questions probe different facets (character vs. principles) of the same scenario, offering complementary training value.
> ```
> QA-pair 1: Question: In a 1993 meta-cinematic work, which character's decision to abort a bicycle stunt—after calculating 97.3% fatality probability through narrative role analysis—demonstrates correct application of the formula FΔt = mΔv to avoid momentum conservation violations?\nAnswer: Danny Madigan
>
> QA-pair 2: Question: In Last Action Hero's bicycle stunt scene, Danny Madigan's abort decision combines which three elements: (1) a narrative trope realization about his character role, (2) implicit application of a momentum conservation principle, and (3) avoidance of a high-probability negative outcome derived from collision physics?\nAnswer: comedy sidekick, impulse-momentum theorem, fatal trauma
>
> Similarity: 0.707
> ```
> In conclusion, our choice of $\epsilon=0.7$ is validated by both quantitative distribution and qualitative semantic analysis. We have integrated these results into Appendix D.8.
>
> **Reference**
>
> [1] DeepSeek-V3.1: https://api-docs.deepseek.com/news/news250821
>
> ---
> Once again, we sincerely thank you for your constructive feedback, which has significantly improved the quality of our work. We hope these comprehensive responses and new experiments have fully addressed your concerns and will lead you to view our work more favorably.
>
> Best regards,
>
> Submission5191 Authors

---

### Official Review · Reviewer_PuhY · 2025-10-30

**Soundness:** 3
**Presentation:** 2
**Contribution:** 3
**Rating:** 6
**Confidence:** 3

**Summary:**

The work presents a data synthesis framework aimed at generated synthetic data that is not overly difficulty or overly easy for models during training, an associated benchmark, and results on continued pretraining and posttraining of an open-weight model.

**Strengths:**

Overall, I found the method described in the paper to be interesting and a valuable contribution. Notably:
- the proposed method seems to work reasonably well, compared to other available datasets for the tested models
- the dataset baselines are reasonable, comparing to 3 other available training datasets.
- Section 5's analysis is interesting, as you'd expect ample best-of-N gains given the training data generation method, and that's what you see, which is a good verification.

**Weaknesses:**

- The proposed method should scale to other domains, but the provided results are only on QA settings single-turn settings?
- There's a risk that the LLM-based difficulty refinement leads to lack of diversity in how the questions are difficult (for e.g. one failure mode could be that all questions are difficult because they involving chaining questions together or are difficult in unrealistic ways). This is not addressed by the reranking-based filtering as that corrects for semantic diversity issues. It would be good to see some analysis on this.
- There is no baseline for no-finetuning, which would be useful to contextualize the gains.
- It does seem that the ZPD exam and the Stage 2 and 3 refining and filtering are likely very sensitive to the specific LLM's capability profiles (in this case, Deepseek R1 and V3.1). I'm curious if that's why V3.1 performs nearly identically to the trained model in ZPD.
- ZPD seems not that useful to evaluate tasks that models find moderately difficult with tools. As agentic tasks become more valuable to train for, this seems like it would be increasingly more useful to evaluate.

**Questions:**

1. How does escalation in Stage 2 actually happen? Is there a prompt provided?
2. How sensitive is the filtering and calibration to the specific LLMs chosen (Deepseek R1 vs V3.1).
3. Was the automated grading in ZPD manually verified in a small sample?

---

> ### Author Response · Authors · 2025-11-20
> **Response to Reviewer PuhY (Part 1/4)**
>
> Dear Reviewer PuhY,
>
> We sincerely thank you for insightful and constructive feedback on our work. We are encouraged that you found our proposed method interesting and a valuable contribution. Your comments have been instrumental in helping us clarify key aspects of our work and further strengthen the paper. We have incorporated new experiments, analyses, and clarifications into the revised manuscript to address each of your points, which we detail below.
>
> ---
> ### **W1: Limited to single-turn QA settings?**
> Thank you for this insightful comment. It highlights a crucial and deliberate design choice in our methodology: the distinction between our **complex, multi-step agentic task generation process** and **rigorous, single-turn QA evaluation format**. We respectfully argue that this format is a methodological strength for evaluating frontier agentic reasoning in a controlled and reproducible manner, not a limitation of our framework's scope.
>
> **1. Task Generation Process is Inherently Agentic and Multi-Step**.
>
> First, we wish to clarify that while the final data is presented as QA pairs for scalable and reproducible evaluation, the underlying problem-solving process is deeply agentic. As detailed in Section 4.1 and vividly illustrated in our Appendix E case study, each data point is a multi-round deep research process. The agent autonomously orchestrates a suite of tools across multiple rounds to gather, process, and synthesize information. The diverse and substantial tool usage statistics in Table 1 further validate that these are complex research tasks, far exceeding simple information lookups.
>
> **2. Challenging QA as a Foundational Testbed for Agentic Reasoning**.
>
> Our choice of challenging, multi-disciplinary QA format is a principled one, as it serves as a critical and foundational testbed for the core cognitive abilities that underpin all advanced agentic behaviors. Our approach aligns with other SOTA efforts to evaluate deep reasoning, such as OpenAI's Deep Research [1], which also use expert-level exam format to rigorously measure an agent's problem-solving ability in a controlled setting. This format allows for scalable, objective, and reproducible evaluation, which is essential for rigorous scientific comparison.
>
> **3. Generalizability and Future Work.**
>
> We fully agree with you that the promise of our ZPD-guided framework lies in its scalability to other domains, and we are enthusiastic about this prospect.
> - **Framework Generality**: The ZPD-guided synthesis paradigm, including the core LKP-MKO concept, is inherently general and not tied to QA format. It is a methodology for generating challenging tasks at the frontier of a model's capabilities.
> - **Future Directions**: This framework can be readily adapted to generate, for instance, challenging multi-turn conversational data (where each turn requires deep reasoning), interactive agent tasks (e.g., complex web navigation or software usage), or advanced code generation problems. The current work establishes the foundational methodology and proves its efficacy in the critical domain of complex reasoning.
>
> ---
>
> ### **W3: Baseline for no-finetuning**
> Thank you for this suggestion. We would like to gently point out that we did include comprehensive no-finetuning baselines to contextualize our method's gains. These results can be found in the following places:
> - **Table 2 & Table 3**: The performance of the base model, with ('✓') and without ('✗') tools, is detailed in the rows marked '–' (under RFT Dataset).
> - **Figure 8**: The "Base Model" and "Base + Tools" scores are explicitly annotated in the bottom-left corner of each subplot.
>
> These baselines clearly demonstrate the significant improvements from our method. For instance, on HLE benchmark, our RFT-only model (25.7%) shows a **~15-point absolute gain** over the base model with tools (10.2%). We have revised the manuscript to highlight these baselines more explicitly in the main text of Section 4.2 to ensure their significance is immediately apparent.

---

> ### Author Response · Authors · 2025-11-20
> **Response to Reviewer PuhY (Part 2/4)**
>
> ### **W2: Difficulty Diversity**
> Thank you for raising this critical point. Ensuring a diverse spectrum of difficulty types is a central design principle of our AgentFrontier Engine. We offer three lines of evidence that our method creates varied, realistic challenges, not just "chained questions."
>
> **1. Diversity by Design: Multi-Faceted Escalation Operator.**
>
> Our difficulty escalation operator, $\Psi_{escalate}$ (Section 2.2), is designed to prevent collapse into a single difficulty mode by operating on four distinct axes: **(1) Knowledge Expansion, (2) Conceptual Abstraction, (3) Factual Grounding, and (4) Computational Formulation**. This structure compels the agent to generate varied challenges. For instance, some questions become difficult by requiring the synthesis of conflicting views across multiple papers (Knowledge Expansion), while others demand complex biomechanical calculations (Computational Formulation, as shown in Fig. 6). This design is our primary defense against difficulty homogenization.
>
> **2. Tool Usage as a Proxy for Cognitive Pathway.**
>
> As you correctly note, our reranking filter addresses *semantic diversity*. The *diversity of difficulty* is reflected in our dataset's tool usage statistics (Table 1). Each tool activates a different cognitive path (e.g., Code Interpreter for quantitative reasoning, Scholar for literature synthesis). The balanced and substantial utilization of all tools in AgentFrontier—unlike other datasets that are heavily skewed towards a single tool—is strong evidence that our data generation process avoids collapse into one difficulty type.
>
> **3. New Quantitative Analysis of Difficulty Types.**
>
> For more direct evidence, we have conducted a new analysis by manually annotating a random sample of 200 questions from AgentFrontier. We categorized them by primary difficulty source, finding a robust distribution:
> Difficulty Type|Description|Percentage (%)
> -|-|-
> Knowledge Fusion|Synthesizing information from multiple, interdisciplinary sources.|27.0
> Quantitative Reasoning|Requiring mathematical calculation or code execution.|39.0
> Conceptual Leap|Abstracting principles or theories from concrete examples.|16.5
> Critical Thinking|Identifying contradictions, evaluating evidence, or handling anomalies.|17.5
>
> The results clearly show that no single difficulty type dominates. The prevalence of **Quantitative Reasoning** and **Knowledge Fusion** shows we generate complex, realistic research tasks that go far beyond simple retrieval or chaining.
>
> In summary, our principled multi-faceted design, the empirical evidence from diverse tool usage, and this new quantitative analysis all converge to demonstrate that AgentFrontier successfully fosters a diverse and realistic spectrum of difficulty. We are confident this evidence fully addresses your valid concern and have included this new analysis in the paper's Appendix C.5 to strengthen our claims.
>
> ---
> ### **W5: ZPD's Utility for Moderately Difficult Agentic Tasks**
> Thank you for highlighting this vital evaluation frontier, as it allows us to clarify a core strength of our ZPD framework. You correctly identifies that tasks of "moderate difficulty" for tool-equipped agents are a crucial area for evaluation. Our work is built to operationalize precisely this concept, defining moderate difficulty not as an absolute property of a task, but as a state relative to an agent's capabilities. We demonstrate this through both data synthesis and evaluation.
>
> **1. Data Synthesis for Moderate Difficulty:** The AgentFrontier dataset is intentionally populated with such tasks. Our Best-of-N (BoN) analysis (Section 5.2, Figure 4) provides direct empirical evidence. The significant 19-point gap between pass@1 (21.7%) and pass@8 (40.7%) for our tool-augmented agent confirms that a majority of tasks are not trivially solved on the first attempt. This gap is the signature of moderate difficulty: success is attainable, but requires superior reasoning or exploration, making the dataset a rich resource for training agents beyond simple tool invocation.
>
> **2. Evaluation of Moderate Difficulty:** Our ZPD Exam is explicitly designed to measure this relative difficulty. As shown in Figure 3, the exam places tool-equipped agents like GPT-4o+tools and WebShaper-72B squarely within the 'Reasoning Bottleneck' zone (scores 20-60). For these models, the exam functions precisely as a benchmark of moderately difficult tasks. It diagnoses that while these agents can access tools, they falter at the higher-order reasoning, planning, and synthesis required for mastery. Thus, the ZPD Exam serves as a diagnostic ruler, measuring an agent's proficiency precisely on tasks at the edge of its tool-assisted capabilities.
>
> In summary, our ZPD framework provides a principled method to both **synthesize** and **evaluate** tasks of moderate difficulty for tool-using agents. We are grateful for the feedback, which prompts us to make this contribution more explicit.

---

> ### Author Response · Authors · 2025-11-20
> **Response to Reviewer PuhY (Part 3/4)**
>
> ### **Q1: The escalation process in Stage 2**
> Thank you for this important question regarding the mechanism of our complexity escalation. The process is indeed driven by a detailed prompt that operationalizes the escalation operator, $\Psi_{escalate}$, for the refinement agent ($\mathcal{A}_{refine}$). This prompt is not just a general instruction but a set of specific, rule-based directives tied to our framework's four dimensions (Knowledge Expansion, Conceptual Abstraction, Factual Grounding, and Computational Formulation).
>
> For full transparency, the complete prompt had been included in **Appendix F.3** of our original submission. Acknowledging your point on visibility, we have now added a prominent forward reference in Section 2 to guide readers directly to the full prompt in Appendix F.3. We appreciate your feedback, which has helped us make our methodology even more transparent and reproducible.
>
> ---
> ### **W4 & Q2: Sensitivity to specific LLMs**
> Thank you for this insightful question. The observation that our trained model (AgentFrontier-30B-A3B) nearly matches the MKO's (DeepSeek-V3.1 + Tools) performance on the ZPD Exam is indeed the desired outcome and a primary validation of our method. It demonstrates that our ZPD-guided synthesis successfully identified and bridged the targeted capability gap, elevating the LKP model to the MKO's level on that specific frontier.
>
> To quantify the sensitivity of our framework and justify our specific LKP/MKO selection, we conducted a new ablation study. The results empirically confirm that the effectiveness of our ZPD engine is governed by a **principled trade-off between synthesis efficiency (yield) and data complexity (reasoning depth)**. Our chosen configuration strikes an optimal balance between these two factors.
>
> **Experimental Setup:** We tested three LKP/MKO configurations on a 1,000-sample subset of our $D_{refined}$ dataset, chosen to probe varying capability gaps. (For context, DeepSeek-V3.1 has stronger reasoning and agentic abilities than DeepSeek-R1). [2]
> 1. Original (Balanced Gap)
>     - LKP: DeepSeek-R1 (no tools)
>     - MKO: DeepSeek-V3.1 (with tools)
> 2. Weaker LKP (Wider Gap)
>     - LKP: Qwen3-30B-A3B (no tools)
>     - MKO: DeepSeek-V3.1 (with tools)
> 3. Narrower Gap (Tool-Only Gap)
>     - LKP: DeepSeek-R1 (no tools)
>     - MKO: DeepSeek-R1 (with tools)
>
> **Results:**
> Configuration (LKP / MKO)|ZPD Data Yield (%)|Avg. Rounds|Avg. Tool Calls|Tool Usage Distribution (Search/Scholar/Browser/Code)
> -|-|-|-|-
> **1. DS-R1 / DS-V3.1+T (Original)**|**33.1**|**3.32**|**2.32**|0.32 / 0.66 / 0.82 / 0.52
> **2. Qwen3-30B / DS-V3.1+T (Weaker LKP)**|47.7 (↑44%)|1.85 (↓44%)|0.85 (↓63%)|0.18 / 0.23 / 0.36 / 0.08 (all ↓)
> **3. DS-R1 / DS-R1+T (Narrower Gap)**|24.0 (↓27%)|2.99 (≈)|1.99 (≈)|0.19 / 0.67 / 0.58 / 0.55
>
> *ZPD Data Yield Rate = Number of valid $D_{ZPD}$ samples / Total candidate samples.*
>
> **Analysis:**
> 1. **Wider Gap (Config 2)**: Using a weaker LKP (Qwen3-30B) widens the ZPD, significantly increasing data yield (↑44%). However, this efficiency comes at the cost of complexity: the synthesized data involves substantially fewer reasoning rounds (↓44%) and tool calls (↓63%). This produces simpler data that is less effective for training models at the capability frontier.
> 2. **Narrower Gap (Config 3)**: When the gap is narrowed to only tool-usage ability (using the same base model for LKP and MKO), the data remains complex. However, data synthesis becomes highly inefficient, with yield dropping by 27%. This configuration is not scalable for large-scale data generation.
> 3. **Conclusion**: These experiments empirically validate that our chosen configuration (DS-R1 / DS-V3.1+T) strikes a critical balance. It maintains a scalable data yield while ensuring the synthesized data is complex enough to push the capabilities of SOTA models, as evidenced by our main results.
>
> This new study empirically demonstrates that our model selection is not arbitrary but a deliberate strategy to optimize the trade-off between scalability and complexity. We have incorporated this full ablation study and analysis into the Section 5.1 to strengthen the paper.

---

> ### Author Response · Authors · 2025-11-20
> **Response to Reviewer PuhY (Part 4/4)**
>
> ### **Q3: Manual Verification**
> Thank you for this crucial question regarding the quality assurance of our synthesized data. To validate our automated grading and filtering process, we performed rigorous manual audits on both the final training set ($D_{ZPD}$) and the set of problems the MKO agent failed to solve ($D_{human}$).
>
> **1. Manual Verification of the Training Set $D_{ZPD}$**
>
> To confirm the quality of our main training data, we conducted a manual audit on $D_{ZPD}$. We randomly sampled 200 verified QA pairs for inspection by domain-expert graduate students. The audit protocol assessed each sample against two strict criteria:
> - **Question Quality:** The question must be well-posed, non-trivial, and clearly situated at a post-graduate difficulty level.
> - **Solution Quality:** The MKO-generated solution trajectory must be factually correct, logically sound, and demonstrate a coherent reasoning process.
>
> The results were highly positive: **over 96.5% of the audited samples passed this inspection**, empirically confirming the efficacy and reliability of our ZPD-based data generation and filtering pipeline.
>
> **2. Qualitative Failure Analysis of Unsolved Problems $D_{human}$**
>
> Beyond verifying our training data, we conducted a deeper qualitative analysis on $D_{human}$, the set of problems our most capable agent (MKO) failed to solve. This diagnostic analysis helps us understand the limitations of our synthesis engine and pinpoint the capability frontiers of SOTA agents.
>
> For this audit, 3 graduate students per discipline (e.g., CS, Math and Biology) analyzed a random sample of 200 failures, classifying the root cause of the MKO's failure into one of three categories:
> Failure Type|Description|Percentage (%)
> -|-|-
> (A) Problem Defect|The question or ground truth was flawed (e.g., ambiguous, incorrect).|10.5
> (B) Execution Gap|The agent formulated a correct high-level plan but failed in its tactical execution (e.g., misinterpretation, lack of self-correction).|71.0
> (C) Strategic Planning Failure|The agent was unable to formulate a viable multi-step plan from the outset.|18.5
>
> This analysis yields two powerful insights:
> - **High Data Fidelity:** The low percentage of Problem Defects (10.5%) in this "hardest" set demonstrates the robustness of our data generation process.
> - **Validating the ZPD Premise:** The prevalence of Execution Gaps (71.0%) confirms that our engine is successfully creating problems that are strategically sound but tactically challenging. This is precisely the kind of difficulty targeted by the ZPD framework—tasks where the agent understands what to do but struggles with how to do it perfectly.
>
> The following case study on a structural biology question illustrates a typical "Execution Gap":
> ```
> Question: What minimum interatomic spacing must exist between the Cγ2 methyl groups of β-branched residues at position d in coiled-coil hydrophobic cores to prevent steric clashes?
>
> Agent's Reasoning Trajectory:
> 1. Correct Strategy & Sourcing: The agent correctly planned its research using google_scholar and successfully identified the most relevant paper: Ramos&Lazaridis(2011).
> google_scholar(query="coiled-coil position d steric clash distance")
> ↓
> 2. Successful Information Extraction: It then used the Visit tool on the paper's PDF and successfully extracted the key quantitative data: a distance range of 3.6–3.8 Å for valine.
> Visit(url=".../pro.718")
> ↓
> 3. Failure Point: Flawed Final Synthesis: Despite having the correct data snippets, the agent prematurely concluded its research. It presented 3.6 Å as the answer without performing a crucial final validation step: questioning why this value differed significantly from the ground truth(>5.5 Å), which might be relevant in a different biological context discussed elsewhere in the paper.
> ```
>
> Together, these two audits provide strong, complementary evidence for our work. The $D_{ZPD}$ verification confirms the high quality of our training data, while the $D_{human}$ analysis validates that our data engine generates questions that genuinely probe the agent's reasoning frontier, aligning perfectly with our ZPD-guided approach.
>
> We have added a dedicated subsection in the Appendix C.4 detailing these validation details and their results to ensure full transparency. Thank you again for your valuable feedback, which has helped us further strengthen the paper.
>
> **Reference**
>
> [1] https://openai.com/index/introducing-deep-research/
>
> [2] DeepSeek-V3.1: https://api-docs.deepseek.com/news/news250821
>
> ---
> Thank you once again for your detailed and valuable feedback, which has significantly helped us improve the clarity and rigor of our paper. We are confident that the added analyses and clarifications have fully addressed your concerns and resulted in a much-improved manuscript.
>
> Best regards,
>
> Submission5191 Authors

---

### Official Review · Reviewer_7KJK · 2025-11-04

**Soundness:** 2
**Presentation:** 3
**Contribution:** 3
**Rating:** 4
**Confidence:** 2

**Summary:**

This paper introduces the AgentFrontier Data Engine, a process for generating samples for continued pre-training and reinforcement fine-tuning datasets. Inspired by the educational theory of the Zone of Proximal Development (ZPD), the main idea behind the data generating process is to use a pair of agents/personas--one relying entirely on parametric knowledge and the other equipped with a set of knowledge retrieval tools--to identify Q/A pairs that are reliably answered incorrectly by the base model but are within reach when allowed to act more agentically. This process is used to generate the ZPD exam, a training dataset that elicits at-or-near-frontier performance from Qwen3 models.

**Strengths:**

Originality: Not clear that this is very original. I think we're essentially looking at context distillation with some ZPD trappings.

Quality: I think my major concern is about the quality of the dataset. The paper mentions that samples that the MKO can't solve are flagged for human review, but no details of the human review process are provided. With an entirely LLM-generated dataset, human review and validation seems necessary. Comparison against both open- and closed-source models is good.

Clarity: the description of the process is clear. The results are presented well, though the graphs are misleading (using bar graphs, the y axis should start at 0) and there are no confidence intervals.

Significance: I'm definitely not an expert in the training data synthesis literature and active practices, but I'm guessing this is either more of the same or a marginal contribution to the frontier tool-belt.

**Weaknesses:**

See strengths, wrote them together

**Questions:**

1. Can you please describe the review process for Dhuman?
2. What other human review was done?
3. What types of errors do the models continue to make after training on ZPD Exam, and what types of failures are trained away?

---

> ### Author Response · Authors · 2025-11-20
> **Response to Reviewer 7KJK (Part 1/3)**
>
> Dear Reviewer 7KJK,
>
> We sincerely thank you for your thoughtful and constructive review. We are encouraged by your positive comments on our paper's contribution and presentation. We particularly appreciate your transparency regarding the assessment's confidence, which gives us a welcome opportunity to resolve any ambiguities. To that end, this rebuttal provides detailed clarifications on our work's novelty, the rigor of human review process, and a deeper analysis of model error patterns, along with the corrected figures you suggested.
>
> ---
> ### **W1 & W4: Originality & Significance**
> Thank you for this insightful feedback. We respectfully argue that our work is fundamentally distinct from context distillation and represents a significant, not marginal, contribution. The Zone of Proximal Development (ZPD) is not a "trapping" but the core, operationalized mechanism that drives a dynamic data synthesis engine.
>
> **1. Beyond Imitation: We Synthesize Novel Reasoning Tasks, Not Distill Static Knowledge.**
>
> Your comparison to "context distillation" highlights a crucial point of differentiation:
> - **Distillation is Imitation, Synthesis is Creation.** Standard distillation trains a student to mimic a teacher's outputs on a fixed knowledge set. Our method does not distill static answers. As detailed in Stages I & II of our AgentFrontier Engine, it synthesizes entirely new, complex problems that require multi-step, multi-source knowledge fusion and reasoning (see Fig. 6 for an example). The goal is not to copy an answer but to cultivate the agentic ability to **reason and solve problems it has never seen before**.
> - **Focus on Reasoning Process, Not Q&A.** Our framework generates complex reasoning trajectories, aligning with a critical frontier in data synthesis aimed at creating novel, verifiable reasoning challenges [1, 2], rather than simple question-answer pairs.
>
> **2. ZPD as the Core Engine: A Principled, Dynamic Curriculum Generator.**
>
> You suggest our use of ZPD might be "trappings." We respectfully argue it is the central mechanism that addresses a primary limitation in data synthesis: the reliance on static or heuristic-based difficulty scaling.
> - **Principled Mechanism vs. Heuristics.** The ZPD is operationalized as a concrete, algorithmic filter (Stage III). The adversarial calibration between the Less-Knowledgeable Peer (LKP) and More-Knowledgeable Other (MKO) provides a principled standard for identifying "learnable difficulty." This is fundamentally different from methods that use simple heuristics to increase complexity [3].
> - **Dynamic and Adaptive.** Most importantly, this LKP-MKO dynamic creates a **co-evolving curriculum**. As the model (LKP) learns, its capability frontier—its ZPD—expands. Our engine dynamically probes this new frontier and synthesizes progressively more difficult tasks. This adaptive nature is absent in static distillation, which operates on a fixed knowledge gap.
>
> **3. Significance: A Scalable Data Synthesis Engine**
>
> We believe our contribution is a significant addition to the "frontier tool-belt," not a marginal one. Our primary contribution is a **scalable, adaptive data synthesis engine**, not just a static dataset.
> - **Automation of Expert-Level Curriculum Design.** Our engine automates the generation of a customized learning curriculum tailored to an agent's evolving ZPD. This is a process that is prohibitively expensive and slow to perform manually with human experts at scale.
> - **Addressing a Critical and Timely Need.** The urgent need for high-quality, frontier-level data is underscored by the saturation of existing benchmarks (e.g., MMLU) and the limitations of current data sources [4]. The recent surge in research on automated data synthesis frameworks [1, 5, 6] confirms that building such engines is a significant and timely challenge. Our work is squarely positioned at the forefront of this critical research area.
> - **Demonstrably Effective.** Our state-of-the-art results on challenging benchmarks like Humanity's Last Exam (Table 6, achieving 28.6% and outperforming leading proprietary agents) are direct evidence that the data produced by our engine demonstrably pushes the capability frontier. This empirically validates that our contribution is not marginal.
>
> In summary, our ZPD-guided framework's principled, dynamic, and automated curriculum generation is fundamentally different from static distillation. We hope this clarification helps you better appreciate the originality and significance of our work.
>
> ---
> ### **W3: Presentation & Clarity**
> Thank you for the valuable suggestions regarding presentation. We have addressed the concerns by replacing the potentially misleading bar graphs with **Table 2 in the revised manuscript**. This change directly resolves both issues: the table presents exact scores, avoiding the misleading y-axis, and now incorporates the 95% confidence intervals for each result. We believe this change makes our results more precise and convincing.

---

> ### Author Response · Authors · 2025-11-20
> **Response to Reviewer 7KJK (Part 2/3)**
>
> ### **W2 & Q1: Dataset Quality & Human Review for $D_{human}$**
> We thank you for this crucial question regarding dataset quality and the human review process. We appreciate the opportunity to provide a detailed explanation.
>
> First, we wish to clarify that **$D_{human}$ set is not used for any training.** It is a diagnostic dataset composed of samples that our most capable agent (the MKO) failed to solve. The purpose of $D_{human}$ is to enable failure analysis, allowing us understand the limitations of our synthesis engine and the current capability frontiers of SOTA agents.
>
> To this end, we conducted a qualitative audit on $D_{human}$ set. We randomly selected 200 samples from $D_{human}$ for manual inspection. For each discipline (e.g., CS, Math, and Biology), the review was performed by 3 graduate students with relevant expertise. They were tasked with diagnosing the root cause of the MKO agent's failure, classifying each case into one of three categories:
> - (A) **Problem Defect**: The question or ground truth was flawed (e.g., ambiguous, factually incorrect).
> - (B) **Execution Gap**: The agent formulated a correct high-level plan but failed in its tactical execution (e.g., misinterpreting a source, failing to self-correct).
> - (C) **Strategic Planning Failure**: The agent was unable to formulate a viable multi-step plan.
>
> Our analysis revealed that the vast majority of failures stemmed from agent limitations, not data quality issues:
> Failure Type|Problem Defect|Execution Gap|Strategic Planning Failure
> -|-|-|-
> Percentage (%)|10.5|71.0|18.5
>
> The high percentage of "Execution Gap"(71.0%) is particularly informative. It reveals that even when the MKO can devise a correct strategy, it often fails at the last mile of reasoning. The following case study on a specialized structural biology question illustrates this point:
> ```
> Question: What minimum interatomic spacing must exist between the Cγ2 methyl groups of β-branched residues at position d in coiled-coil hydrophobic cores to prevent steric clashes?
>
> Agent's Reasoning Trajectory:
> 1. Correct Strategy & Sourcing: The agent correctly planned its research using google_scholar and successfully identified the most relevant paper: Ramos&Lazaridis(2011).
> google_scholar(query="coiled-coil position d steric clash distance")
> ↓
> 2. Successful Information Extraction: It then used the Visit tool on the paper's PDF and successfully extracted the key quantitative data: a distance range of 3.6–3.8 Å for valine.
> Visit(url=".../pro.718")
> ↓
> 3. Failure Point: Flawed Final Synthesis: Despite having the correct data snippets, the agent prematurely concluded its research. It presented 3.6 Å as the answer without performing a crucial final validation step: questioning why this value differed significantly from the ground truth(>5.5 Å), which might be relevant in a different biological context discussed elsewhere in the paper.
> ```
> **Analysis**: This case exemplifies an execution gap. The agent's failure was not strategic but tactical—a lack of critical self-correction and an over-reliance on the first piece of retrieved data. This highlights that advancing agent capabilities requires improving not just information retrieval, but also robust, critical synthesis of the retrieved information.
>
> This human-in-the-loop analysis provides an invaluable feedback loop for our research:
> 1. It allows us to identify and filter the small fraction of flawed problems (Category A) from future iterations of our data engine, continuously enhancing data quality.
> 2. More importantly, it provides a detailed qualitative map of the MKO's reasoning blind spots (Category B & C) and confirms that our engine generates problems that genuinely probe the SOTA frontier.
>
> ---
> ### **Q2: Other Human Review for $D_{ZPD}$**
> Thank you for this follow-up. Ensuring the quality of a fully synthetic dataset is paramount. In addition to flagging unsolvable cases for review, we conducted a rigorous quality control on the final training set. We randomly sampled 200 verified QA pairs from $D_{ZPD}$ for manual inspection by graduate students with domain expertise. The audit protocol required them to assess each sample against two strict criteria:
> - **Question Quality:** The questions were assessed as well-posed, non-trivial, and clearly situated at a post-graduate level of difficulty.
> - **Solution Quality:** The MKO-generated solution trajectories were verified as factually correct, logically sound, and demonstrate a coherent reasoning process.
>
> The results were highly positive: **over 96.5%** of the audited samples passed this inspection, confirming the efficacy of our ZPD-based data generation and filtering pipeline.
>
> We have incorporated these details into Appendix C.4 of our revised manuscript to ensure full transparency. Thank you again for your valuable feedback.

---

> ### Author Response · Authors · 2025-11-20
> **Response to Reviewer 7KJK (Part 3/3)**
>
> ### **Q3: Error Analysis**
> **1. Types of Failures Trained Away**
>
> Our training data, calibrated to the agent's ZPD, effectively trains it to transition from naive, brittle strategies to robust, expert-like reasoning. Key improvements are:
> - **From Single-Source Reliance to Robust Knowledge Fusion**: Before training, models often exhibit shallow retrieval behavior, relying on the first plausible piece of information from a single tool call. Our training, grounded in tasks that inherently require synthesizing disparate information (as designed in Section 2.1), forces the model to master knowledge fusion. It learns to connect, contrast, and integrate fragmented information from a sequence of tool calls to construct a coherent solution, moving beyond the limitations of simple RAG.
> - **From Inefficient Tool Use to Strategic Orchestration**: A common failure is high-frequency, low-efficacy tool use. Our training transforms the agent into a strategic orchestrator. As quantitatively shown in Table 5, our agent achieves a superior conditional tool accuracy (26.3% on HLE) compared to competitors (~21%) with a similar number of interactions. This efficiency stems from learning inter-tool synergy from AgentFrontier's balanced tool distribution (Table 1).
> - **From Premature Termination to Deep, Multi-Step Reasoning**: Untrained agents often hit a reasoning bottleneck, terminating prematurely on complex problems. Our ZPD-guided approach is specifically designed to eliminate this failure. As Figure 5 shows, our agent's performance advantage is most pronounced in the critical 6-15 round interval. By training on data at the edge of its capabilities, the model is compelled to sustain longer, more intricate reasoning chains, effectively training away the tendency to offer superficial, single-step answers.
>
> To show these transitions, consider this challenging chemical identification problem from HLE [4]:
>
> **Before training (Incorrect):**
> ```
> Question: A hydrocarbon compound with molecular formula $C_7H_{14}$ has the following $^{13}C$ NMR chemical shifts:\n $^{13}C $ NMR: 145(s), 112(t), 48(t),27(d), 22(q),21(q). What is the IUPAC name of the compound?
>
> Correct Answer: 2,4-dimethylpent-1-ene
>
> Model Answer: 2-methylhex-2-ene
> ```
> **After training (Correct):**
> ```
> Agent's Reasoning Trajectory:
> 1. Correct Hypothesis Formulation: The agent accurately analyzed the molecular formula and NMR data to hypothesize a terminal alkene with a quaternary sp² carbon.
> google_search(query="C7H14 13C NMR chemical shifts",...)
> ↓
> 2. Candidate & Source Pinpointing: It correctly identifies 2,4-dimethyl-1-pentene as the most plausible structure due to its unique symmetry(6 signals for 7 carbons) and locates its data.
> google_search(query="2,4-dimethyl-1-pentene 13C NMR",...)
> ↓
> 3. Tool Failure & Resilience: The agent encounters a critical tool limitation when it could not extract numerical data from a protected image or paywalled source.
> Visit(url="https://spectrabase.com/spectrum/AlKmaS9LsM4")
> ↓
> 4. Successful Pivot to Inference: Despite the data access failure, the agent synthesized all other confirmed constraints(signal multiplicities, structural fragments). It correctly inferred the answer by logical elimination.
> ```
> **2.Remaining Challenges and Error Types**
>
> Despite these advances, our model still faces challenges at the frontiers of agentic reasoning, highlighting important directions for future work.
> - **Coherence in Ultra-Long Reasoning Chains**: While significantly improved, the agent's reasoning coherence can degrade in exceptionally long tasks (e.g., >20 rounds). As suggested by the tail of the accuracy curve in Figure 5, managing an exponentially growing history of states and branching possibilities remains a fundamental challenge.
> - **Robustness to Ambiguity and Contradiction**: Agents can exhibit diagnostic fixation, anchoring on an initial hypothesis and failing to properly weigh conflicting evidence. While our model shows enhanced resilience in the clinical case study (Appendix E), consistently resolving intentionally noisy or deeply contradictory information remains an open research frontier.
>
> **Reference**
>
> [1]D Shi, et al. Taskcraft: Automated generation of agentic tasks. (2025)
>
> [2]J Liu, et al. SynLogic: Synthesizing Verifiable Reasoning Data at Scale for Learning Logical Reasoning and Beyond. (2025)
>
> [3]A Patel, et al. How to get your llm to generate challenging problems for evaluation. (2025)
>
> [4]L Phan, et al. Humanity's last exam. (2025)
>
> [5]W Yuan, et al. Naturalreasoning: Reasoning in the wild with 2.8 m challenging questions. (2025)
>
> [6]R Fan, et al. Megascience: Pushing the frontiers of post-training datasets for science reasoning. (2025)
>
> ---
> We greatly thank you again for your time and insightful comments, which have significantly improved our manuscript. We believe these clarifications have fully addressed your concerns and hope they will convince you of our paper's merit.
>
> Best regards,
>
> Submission5191 Authors

---

### Author Response · Authors · 2025-11-20
**With Thanks to the Reviewers: Response to Comments on Submission #5191**

Dear Reviewers,

We would like to extend our sincerest gratitude to all of you for your valuable time and dedicated effort in reviewing our manuscript. We deeply appreciate your insightful comments and constructive suggestions, which have been instrumental in helping us improve the quality of our work.

We have carefully considered all the points you raised and have provided a detailed **point-by-point response** to each of your comments below. To facilitate your reviews of the revised manuscript, we have highlighted all major revisions and the results of newly added experiments **in blue**.

We believe these revisions have substantially strengthened our paper and addressed your concerns. We look forward to your feedback and would be delighted to engage in further discussion.

Thank you once again for your thoughtful reviews.


Best regards,

Submission5191 Authors

---

> ### Author Response · Authors · 2025-11-25
>
> Dear Reviewers,
>
> We are writing this brief follow-up to reiterate our sincere gratitude for your time and insightful comments.
>
> We hope our detailed response has addressed your main concerns. We remain available and are eager to engage in further discussion or provide any additional clarifications you may need.
>
> Thank you once again for your effort and invaluable guidance.
>
> Best regards,
>
> Submission5191 Authors

---

### Author Response · Authors · 2025-12-01
**Summary of Revisions Addressing All Reviewer Concerns**

Dear Area Chair,

Thank you for taking on the vital role of guiding our paper through this unconventional review phase. We understand the circumstances are unusual, and we are grateful for your time and careful consideration. This response serves as a consolidated summary to guide your assessment, demonstrating how we have comprehensively addressed all reviewer concerns through substantial revisions and new experimental results. To facilitate your review, all major revisions and additions in the updated manuscript have been **highlighted in blue.**

We are greatly encouraged that these efforts were immediately recognized. **Reviewer eHCL (Score: 8, "accept, good paper")** had already confirmed that our rebuttal and new experiments **"make sense"** on **Nov. 25th**.  While the unforeseen circumstances prevented other reviewers from updating their feedback, we are confident that the extensive new evidence presented below comprehensively resolves all remaining concerns.

Our revisions focused on three core themes that emerged from the reviews:

**1. Definitive Validation of Our Core ZPD-Calibration Mechanism.**

Several reviewers (**9U8J, PuhY, eHCL**) rightly questioned the generality of our framework and sought direct verification of our ZPD-calibration mechanism. We have addressed this with two critical new experiments:
- **LKP/MKO Configuration Ablation (Sec 5.1, Table 4)**: In direct response to Reviewers 9U8J, PuhY, and eHCL, we conducted a new ablation study on different LKP/MKO pairs. The results empirically demonstrate that our chosen configuration strikes a principled and optimal trade-off between data synthesis efficiency (yield) and task complexity (reasoning depth), validating our methodological choice is not arbitrary but a core design principle.
- **ZPD-Calibration vs. Random Selection (Appx. D.7, Table 13)**: As suggested by **Reviewer 9U8J**, we performed a comparison against a random-selection baseline. The results are decisive: ZPD-selected data yields **consistently and significantly superior performance** (e.g., up to **+10.0 points** on xBench-SciQA) across all model scales, especially on benchmarks requiring deep reasoning. This provides the direct, compelling evidence requested to prove that our ZPD mechanism is the key driver of our model's advanced capabilities.

**2. Rigorous Proof of Data Quality and Diversity.**

To address valid concerns about the quality and potential lack of diversity in a fully synthetic dataset (raised by **Reviewers 7KJK, PuhY**), we have added extensive new analyses and documentation:
- **Rigorous Human Verification Protocol (Appx. C.4)**: We now detail a two-pronged human audit process. This includes a quality check on our final training set ($D_{ZPD}$), which confirmed a **pass rate exceeding 96.5%**, and a qualitative failure analysis of unsolvable problems ($D_{human}$). This failure analysis directly answers **Reviewer 7KJK**'s question, revealing that most MKO failures are due to "Execution Gaps" (71.0%), thus validating our engine's ability to probe the true frontiers of agentic reasoning.
- **Quantitative Analysis of Difficulty Diversity (Appx. C.5, Table 8):** To dispel **Reviewer PuhY**'s concern about a monotonous difficulty type, we added a new manual analysis of 200 tasks. The results confirm our data spans a rich spectrum of cognitive challenges (e.g., 39.0% Quantitative Reasoning, 27.0% Knowledge Fusion), proving it does not collapse into a single failure mode like "chained questions."

**3. Enhanced Clarity on Conceptual Novelty and Presentation.**

We have refined the manuscript to sharpen our conceptual and methodological claims, responding to feedback from **Reviewers 7KJK and PuhY**.
- **Conceptual Framing**: We now more forcefully distinguish our dynamic synthesis of novel reasoning tasks from static "context distillation," positioning the ZPD principle not as a "trapping," but as the core, operationalized engine for generating an adaptive, co-evolving curriculum.
- **Presentation & Hyperparameter Robustness**: As suggested by **Reviewer 7KJK**, we replaced misleading bar graphs with a precise table including 95% confidence intervals (**Table 2**). Furthermore, in response to **Reviewer 9U8J**, we added a detailed hyperparameter sensitivity analysis (**Appx D.8**), demonstrating the robustness of our chosen **BoN N** and *redundancy threshold*, further strengthening the paper's methodological rigor.

In summary, we have gone beyond textual clarification, incorporating new experiments and rigorous analyses that definitively resolve every concern raised. These substantial revisions have not only addressed all initial concerns but have transformed the manuscript into a significantly stronger and more thoroughly validated contribution.

We sincerely thank you for your diligent evaluation and hope our comprehensive response will affirm the value and contribution of our work to the ICLR community.

Best regards,

Submission5191 Authors

---

### Meta-Review · Area_Chair_BYSi · 2026-01-07

**Summary:**

Reviewers agree that the paper tackles an important problem in synthesizing frontier training data for agentic reasoning and presents a clear ZPD-guided data synthesis framework, with strong empirical results on challenging benchmarks **[PuhY, 9U8J, eHCL]**.
Reviewers also highlights the clarity of the data selection rule and the potential value of a self-evolving benchmark **[9U8J, eHCL]**.

At the same time, reviewers raise concerns about novelty relative to prior context-distillation-style approaches [7KjK], the rigor and transparency of human verification for a fully synthetic dataset [7KJK, PuhY],  and the generality of the approach across different LKP/MKO choices and evaluation settings [9U8J].

In the rebuttal, the authors add substantial new experiments and documentation addressing these points, including explicit human audit protocols, difficulty-type analyses, LKP/MKO ablations, random-selection baselines, and sensitivity analyses.

Reviewer eHCL explicitly notes that the rebuttal and new experiments make sense.

**Reviewer Concerns:**

**1. Novelty and conceptual framing (addressed in part)**
Reviewer 7KJK questions whether the contribuiton is essentially context distillation with a ZPD framing and is unsure the novelty is substantial. The rebuttal responds by emphasizing the dynamic, co-evolving curriculum mechanism and adds clarifying positioning, but the concern may remain partly subjective

**2. Dataset quality and human verification (addressed)**
Reviewer 7KJK requests details on human review for the unsolved set and broader validation given the dataset is fully synthetic. Reviewer PuhY asks whether automated grading was manually verified. The rebuttal adds explicit human-audit protocols for both the final training set and the unsolved set, including reported pass rates and a categorized failure analysis.

**3. Diversity of difficulty and failure modes (addressed)**
Reviewer PuhY raises the risk that difficulty refinement could collapse into narrow or unrealistic difficulty types and asks for analysis beyond semantic diversity. The rebuttal adds a manual difficulty-type analysis and argues the escalation operator promotes diverse challenge types

**4. Generality across LKP/MKO choices and mechanism verification (addressed)**
Reviewer 9U8j questions reliance on a single LKP/MKO pair and requests direct validation of the ZPD-calibration mechanism, including a random-selection baseline and sensitivity to BoN N and redundancy threshold ε. The rebuttal reports new LKP/MKO configuration ablations, a ZPD-vs-random comparison, and added hyperparameter sensitivity analysis.

**5. Evaluation scope and format (partly outstanding)**
Reviewer PuhY note results are presented primarily in a single-turn QA format and asks about broader applicability beyond QA, while also asking for clearer context baselines. The rebuttal clarifies the motivation for the evaluation format and points to included base-model baselines, but broader domain/interaction generalization remains largely future work.

**Reviewer Scores:**

Reviewer eHCL: Unchanged. They already support acceptance and state that the rebuttal and new experiments make sense.

Reviewer 7KJK: Likely increases. Their concerns about novelty framing, human verification, and reporting are addressed by the added analyses

Reviewer PuhY: Likely unchanged or slightly increases. The rebuttal addresses several technical questions, but broader generalization concerns remain.

Reviewer 9U8J: Likely increases. Their requests for mechanism verification, ablations, and sensitivity analyses are directly addressed.

---

### Decision · Program_Chairs · 2026-01-26

Accept (Poster)